# Timing and nature of AMOC recovery across Termination 2 and magnitude of deglacial $CO_2$ change

Emily L. Deaney[1], Stephen Barker[1] & Tina van de Flierdt[2]

Large amplitude variations in atmospheric $CO_2$ were associated with glacial terminations of the Late Pleistocene. Here we provide multiple lines of evidence suggesting that the $\sim 20$ p.p.m.v. overshoot in $CO_2$ at the end of Termination 2 (T2) $\sim 129$ ka was associated with an abrupt ($\leq 400$ year) deepening of Atlantic Meridional Overturning Circulation (AMOC). In contrast to Termination 1 (T1), which was interrupted by the Bølling-Allerød (B-A), AMOC recovery did not occur until the very end of T2, and was characterized by pronounced formation of deep waters in the NW Atlantic. Considering the variable influences of ocean circulation change on atmospheric $CO_2$, we suggest that the net change in $CO_2$ across the last 2 terminations was approximately equal if the transient effects of deglacial oscillations in ocean circulation are taken into account.

[1] School of Earth and Ocean Sciences, Cardiff University, Main Building, Park Place, Cardiff CF10 3AT, UK. [2] Department of Earth Science and Engineering, South Kensington Campus, Imperial College London, London SW7 2AZ, UK. Correspondence and requests for materials should be addressed to S.B. (email: barkers3@cf.ac.uk).

The Late Pleistocene was characterized by large amplitude variations in atmospheric carbon dioxide ($CO_2$) with corresponding changes in temperature and ice volume[1–3]. The overall pattern of this glacial–interglacial (G–IG) variability is of a gradual build-up of ice sheets as atmospheric $CO_2$ decreases over tens of thousands of years, followed by a relatively rapid collapse of ice sheets and rise in $CO_2$ during deglaciation or glacial termination[4]. The last four glacial cycles (spanning the past ~430 kyr) show a relatively uniform saw-tooth pattern of G–IG variability but the amplitude of $CO_2$ change across their terminations is quite variable. For example, the change in atmospheric $CO_2$ across the most recent deglaciation (Termination, T1) was ~80 p.p.m.v. compared with a rise of ~100 p.p.m.v. across T2 (refs 5,6). Most recent attempts to explain the amplitude of G–IG $CO_2$ variability call upon several distinct mechanisms or processes with variable (and often opposing) effects on $CO_2$ (refs 7,8). In principle therefore, variations in the relative timing of such changes across glacial terminations could give rise to differences in the apparent magnitude of deglacial $CO_2$ change.

Changes in ocean circulation are thought to play a critical role in atmospheric $CO_2$ variability[9–11] and the process of deglaciation itself[12–15] yet the timing of such changes across T2 is poorly constrained. By analogy with similar conditions associated with Heinrich Stadial 1 (HS1, ~18–14.6 ka (ref. 15)) during the early part of T1, evidence from North Atlantic marine sediments[16–18], Chinese speleothems[12] and Antarctic ice cores[1,13] has been used to infer that the Atlantic Meridional Overturning Circulation (AMOC) may have been in a weakened and or shallow mode throughout much of T2 (refs 12,13; during a prolonged interval of North Atlantic cold identified as HS11, ~135–129 ka (ref. 19)) with resumption to a deep and warm mode of circulation occurring only later (~124–127 ka) within the penultimate interglacial period, Marine Isotope Stage (MIS) 5e (refs 17,18). Crucially, if this were the case, it would make T2 quite distinct from the most recent termination, which was characterized by two episodes of weakened AMOC (HS1 and the Younger Dryas (YD)), that were interrupted by an interval of invigorated circulation during the Bølling-Allerød (B-A)[20–22].

Recently published records of seawater Nd isotopes (measured on bulk sediment leachates) and sedimentary Pa/Th from ODP Site 1063 retrieved from the NW Atlantic[17] support the notion of a weakened AMOC during HS11 with a reactivation of North Atlantic Deep Water (NADW) formation during MIS 5e. However, the lack of precise age control across T2, makes it difficult to assess the timing of this change with respect to the record of atmospheric $CO_2$. Here we present new records of Nd isotopes measured on fossil fish debris (Methods section) together with benthic foraminiferal $\delta^{13}C$ and $\delta^{18}O$ from the same core (ODP Site 1063; 33.69° N, 57.62° W, 4,584 m water depth; Fig. 1). To further characterize environmental conditions across T2 and, crucially, to allow us to place our records within the chronostratigraphic framework of the ice-core records[23] (Methods section) we also present new high resolution records of planktic foraminiferal $\delta^{18}O$ (measured on *Globorotalia inflata*), planktic foraminiferal faunal abundance, ice rafted debris (IRD) counts, and additional carbonate preservation indices, all measured on the same samples from ODP Site 1063. In addition we present new sortable silt (SS) measurements (Methods section) from ODP Site 983 (60.48 N, 23.68 W, 1,984 m water depth), to assess the strength of deep water overflows (at that site) emanating from the Nordic Seas[24]. Our results suggest that the AMOC was supressed throughout HS11 with an abrupt resumption of northern deep water production ~129 ka and recovery to a modern-like mode of AMOC by ~124 ka. By comparison with T1 (which experienced a relatively early AMOC

recovery associated with the B-A) we conclude that differences in the apparent magnitude of $CO_2$ change across the last two terminations can be explained, at least in part, by differences in the sequence and timing of deglacial events and their relative influence on atmospheric $CO_2$.

## Results

**Surface ocean properties**. Our new records of planktic foraminiferal $\delta^{18}O$, fauna and IRD counts (Fig. 2a) provide information on upper ocean conditions that we exploit for age model development (Methods section). The abundance record of cold adapted species (*Neogloboquadrina pachyderma* plus *Neogloboquadrina incompta*) reveals several intervals of colder surface conditions, which are also reflected by more positive planktic $\delta^{18}O$ values and the presence of IRD. The records suggest that the interval assigned to HS11 experienced the coldest conditions within the period of interest, with the warmest conditions (as identified from the abundance of warm foraminiferal species: *Globigerinoides ruber* plus *Globigerinoides sacculifer*) being attained directly following HS11, during the earliest part of MIS 5e.

We tie the sharp warming at the end of HS11 in our records to the sharp increase in atmospheric $CH_4$ at ~128.7 ka (128.9–128.5 ka) on the ice-core timescale, AICC2012 (ref. 23; Methods section; Fig. 2a). This age estimate is in good agreement with other recent studies. For example, Jiménez-Amat and Zahn[19] derive an age of 128.73 ka for an abrupt warming recorded in the Alboran Sea following HS11 by correlation to an Italian speleothem record. Another recent study based on an alternative age modelling strategy placed the end of HS11 at 130 ± 2 ka (ref. 25). For comparison the 1σ uncertainty of the AICC2012 timescale at 128.7 ka is ± 1.7 kyr (ref. 23). The most precise date attained for the implied shift from weak to strong Asian Monsoon rainfall associated with the end of HS11 is 129.0 ± 0.1 ka, derived from a speleothem collected in Sanbao Cave, China[12].

**Changes in deep-ocean water mass structure**. Modern bottom water at ODP Site 1063 has a Nd isotopic composition (expressed as $\varepsilon_{Nd}$, the deviation of the measured $^{143}Nd/^{144}Nd$ ratio from the chondritic uniform reservoir in parts per 10,000) of ~ − 13, reflecting the exported mixture of deep waters formed in the eastern and western North Atlantic (NADW) and those emanating from the Southern Ocean[26] (Fig. 1). In agreement with ref. 17 we reconstruct significantly less negative values ($\varepsilon_{Nd} > -11$) at the end of MIS 6 and throughout HS11 (Fig. 2b). In the modern ocean, less negative εNd values are characteristic of deep waters emanating from the south but also those derived from wintertime convection in the Nordic Seas (Fig. 1). Thus the record of εNd from ODP Site 1063 cannot be interpreted as a simple proxy for the mixing ratio between northern and southern deep water sources even if the εNd composition of the various deep water end-members remained constant[27]. On the other hand these deep water masses can be differentiated by their very different $\delta^{13}C$ signatures and carbonate ion concentrations, with northern deep water masses being better ventilated (higher $\delta^{13}C$ and $[CO_3^=]$) than their southern counterparts[20,28]. The low values of benthic $\delta^{13}C$ and poor carbonate preservation (low % coarse fraction and high fragmentation) we observe during HS11 (Fig. 2b), in combination with less negative εNd values, therefore suggest an enhanced influence of southern-sourced deep waters (glacial equivalent to modern Antarctic Bottom Water (AABW)) in the abyssal North Atlantic during HS11 relative to today. This is analogous to the millennial-scale cold events of the last glacial cycle[29]. We note that our new record of benthic $\delta^{13}C$ shows

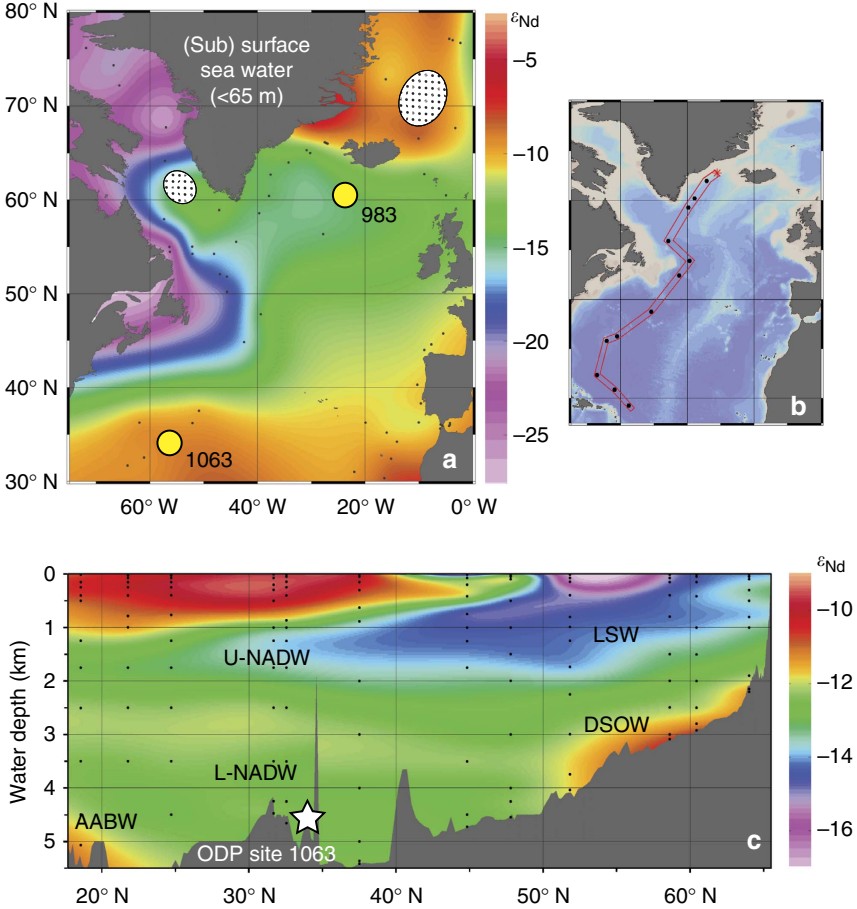

**Figure 1 | Core location and modern seawater εNd.** (**a**) Map of modern near-surface seawater Nd isotopic composition (Methods section). Stippled areas are regions of modern intermediate and deep water production. Locations of ODP sites 983 and 1063 are also indicated. (**b,c**) Station map and seawater Nd isotopic compositions from GEOTRACES section GA02 (ref. 26) highlighting that in the modern ocean only U-NADW (intermediate water) carries a very negative εNd fingerprint, derived from subduction of waters in the Labrador Sea (NW Atlantic Ocean). Overflow waters from the NE Atlantic Ocean, which form the pre-cursor water masses for L-NADW, carry a more radiogenic (higher εNd) fingerprint. DSOW is Denmark Strait Overflow Water. ODP site 983 is bathed by ISOW (not shown), which has a similar εNd composition to DSOW. Map and sections created using the ODV programme (Schlitzer, R., Ocean Data View, http://odv.awi.de, 2016).

considerably more scatter than might be expected for an epifaunal species such as *Cibicidoides wuellerstorfi*, an observation that has also been made for other $\delta^{13}$C records from this region[30] and more widely[18]. While this scatter is difficult to explain, it is possibly due to a fluctuating supply of organic material to the seafloor, occasionally overprinting the bottom water signal. Following previous studies[18] we therefore apply a running mean (3 point) to the benthic $\delta^{13}$C record from ODP Site 1063.

Our records reveal a shift towards higher benthic $\delta^{13}$C and better preservation at the same time as surface ocean warming following HS11 (Fig. 2) suggesting the incursion of better ventilated deep waters to the abyssal North Atlantic likely in response to the renewed penetration of northern-sourced deep waters at this time. Although the absolute value of benthic foraminiferal $\delta^{13}$C can be influenced by whole-ocean changes in $\delta^{13}$C, as well as local effects due to organic matter respiration, the convergence between our new record from ODP Site 1063 (4,584 m) and that from intermediate depth ODP Site 983 (1,984 m)[18] from the relatively remote NE Atlantic (Fig. 3b) suggests that a switch from a glacial to an interglacial-like mode of AMOC[28] occurred at the start of MIS 5e. In combination with a pronounced peak in carbonate preservation at this time, as compared with the latter part of MIS 5e (Fig. 2), these observations are reminiscent of the extreme deepening and

overshoot of the AMOC associated with the B-A during T1 (refs 20,31,32). Importantly our findings confirm that there was no equivalent to the B-A (or YD) during T2 (ref. 33), although there is evidence for a millennial-scale cooling event directly following our inferred overshoot of the AMOC ∼124–125 ka (ref. 34) and supported here by the record of benthic $\delta^{13}$C from ODP Site 1063 (Fig. 2b).

**Changes in the locus of formation of Atlantic deep water.** A deepening of NADW at the onset of MIS 5e could also be inferred from the shift to more negative seawater εNd values at this time[17] (Fig. 2b). However, unlike the equivalent change obtained for the B-A (ref. 21), the very negative εNd values (< −15) attained during early MIS 5e demand a different deep water mass configuration relative to that of the present-day North Atlantic, and a source of bottom water with no modern analogue. At present, NADW represents a predominant mixture of deep waters formed in the NE Atlantic (overflows from the Nordic Seas) and the NW Atlantic (Labrador Sea; Fig. 1). Being colder and saltier than their western counterparts, deep waters produced in the Nordic Seas form so-called Lower- and Middle-NADW (summarized here as L-NADW). Today these waters occupy depths below ∼2,500 m at ODP Site 1063, and are characterized

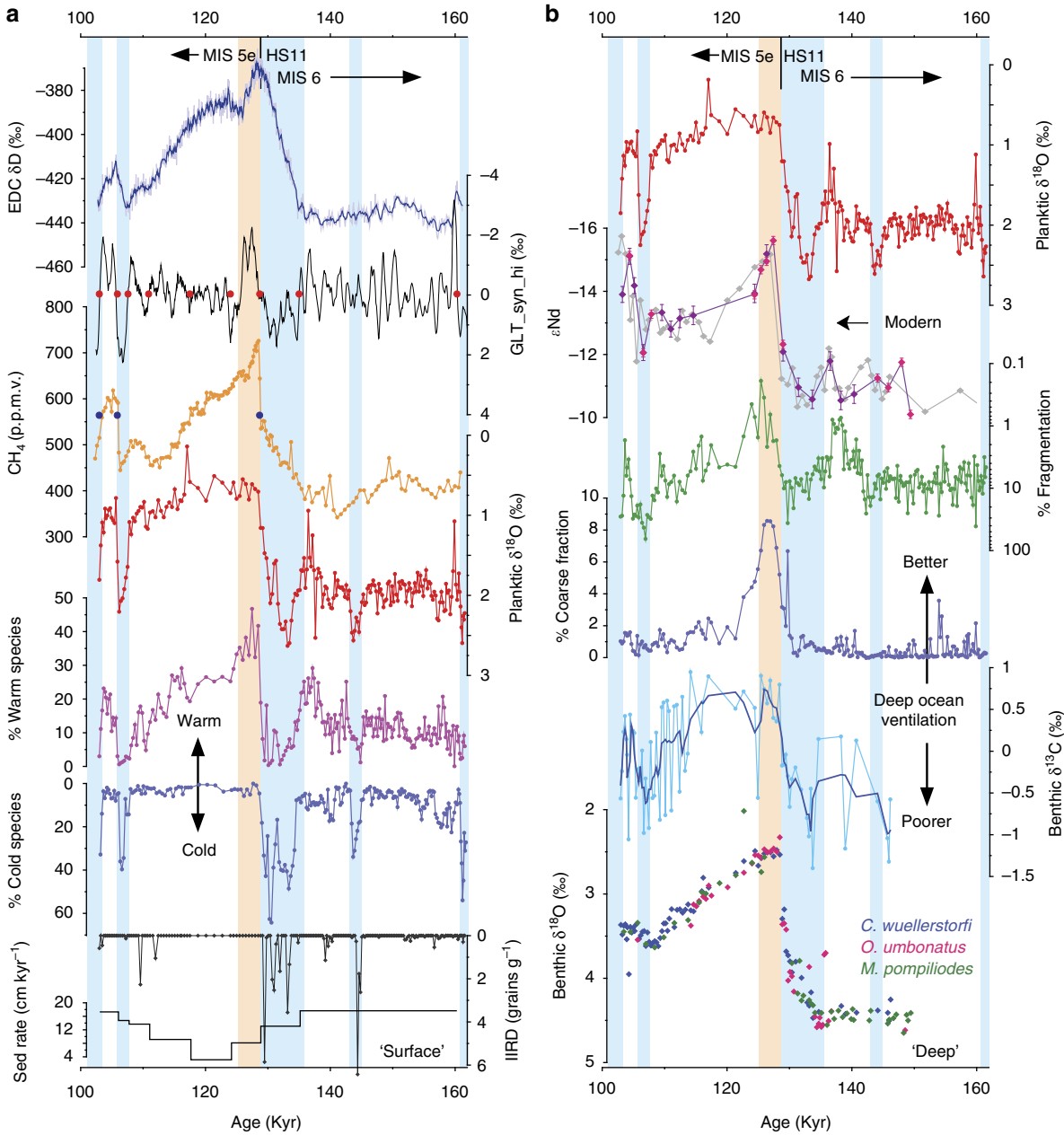

**Figure 2 | Surface and deep records from ODP Site 1063.** (**a**) (from top to bottom) Antarctic ice core temperature proxy, EDC (EPICA Dome C) δD (ref. 1); GL_T_syn_hi, a proxy for anomalous northward heat transport associated with the bipolar seesaw (derived from EDC δD)[13]; atmospheric CH₄ from EDC (ref. 69; all other records are from ODP 1063) planktonic δ¹⁸O measured on *G. inflata*; percentage of warm foraminiferal species (see text); percentage of cold water species; number of IRD grains g⁻¹; implied sedimentation rate. Filled blue and red circles are tuning points between site 1063 and CH₄ and or GL_T_syn_hi, respectively (Methods section). (**b**) (from top to bottom); planktonic δ¹⁸O; seawater $\varepsilon_{Nd}$ (pink symbols were measured by TIMS, purple by MC-ICP-MS, grey are results of ref. 17, error bars are 2σ, arrow is modern value at 1063); percentage of foraminiferal fragmentation; percentage of coarse (>63 μm) fraction; benthic δ¹³C (*C. wuellerstorfi* with 3 point running mean); benthic δ¹⁸O (*C. wuellerstorfi*, *M. pompilioides* (−0.15; Methods section) and *O. umbonatus* (−0.38)). Blue vertical boxes are cold intervals, pink box is our inferred overshoot of the AMOC.

by a Nd isotopic composition of −12.4 ± 0.4 (ref. 26). On the other hand wintertime convection in the Labrador Sea produces relatively warm and fresh Labrador Sea Water (LSW), which forms the upper component of NADW (U-NADW), and carries a Nd isotope fingerprint of $\varepsilon_{Nd} = -14.2 \pm 0.3$ (at its extreme)[26]. At 4,584 m water depth, ODP Site 1063 lies within L-NADW, with only minor influence from colder and fresher AABW, and well outside the density range of LSW (Fig. 1). An obvious way of making the Nd isotopic composition of bottom waters at the site of ODP Site 1063 more negative would be to increase the influence of the very negative $\varepsilon_{Nd}$ surface waters found in the NW

Atlantic (Baffin Bay and the Labrador Coast; Fig. 1). However, simply increasing the (volumetric) contribution of modern LSW to NADW, or increasing the Nd concentration in LSW[17], is not sufficient to explain the negative values we observe at our site during early MIS 5e.

Surface waters with sufficiently negative Nd isotopic compositions ($\varepsilon_{Nd} \ll -14$) are observed north and south of the modern convection areas in the Labrador Sea and it is feasible that convection during early MIS 5e was shifted to a more southerly location compared to today, tapping into surface waters with more negative Nd isotopic composition. It is also possible that

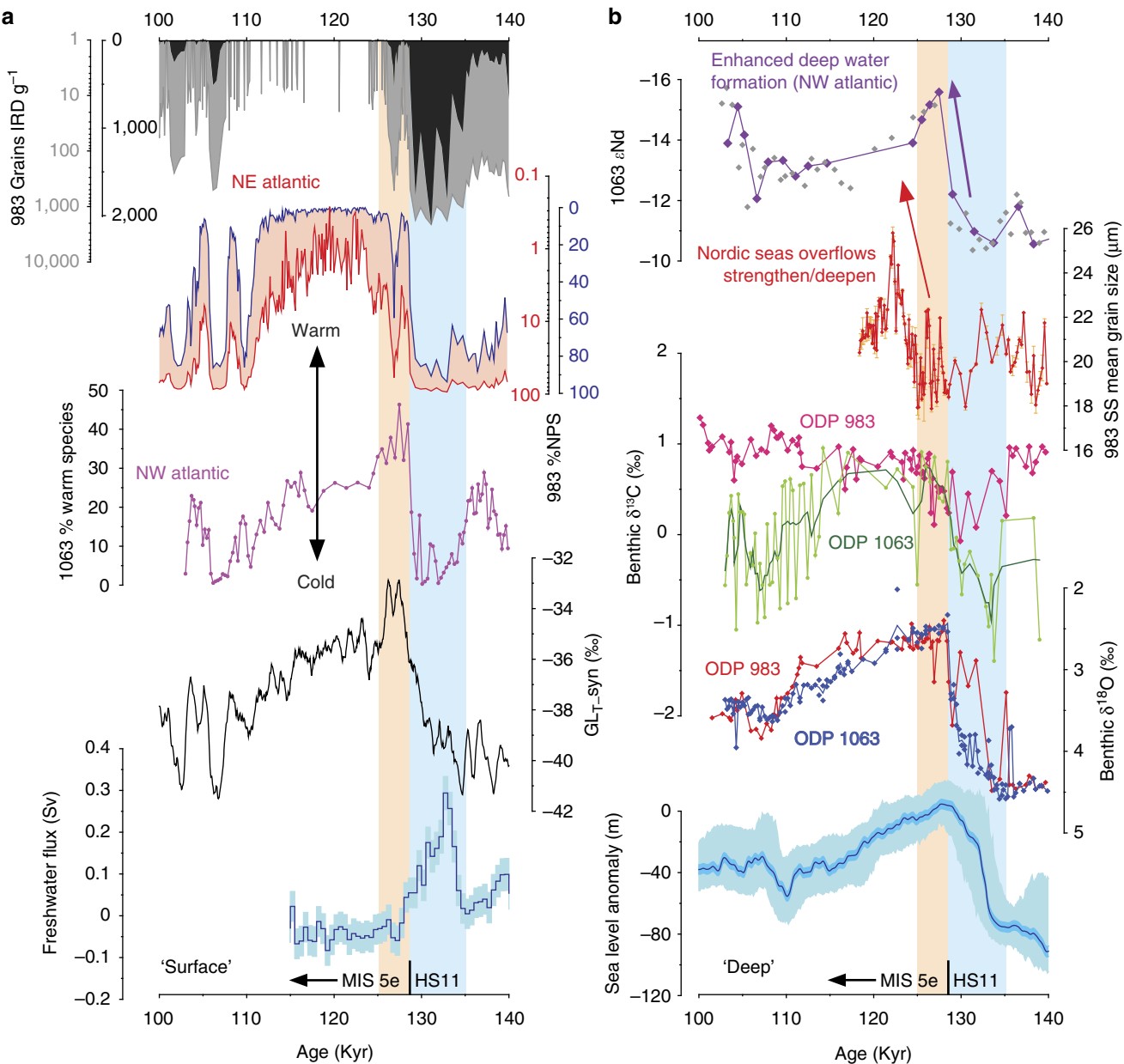

**Figure 3 | Surface and deep ocean properties across MIS 5e. (a)** (from top to bottom) IRD (grains.g$^{-1}$; black and grey curves are on linear and log scales, respectively) and %NPS (reflecting surface temperature, blue and red curves are on linear and log scales, respectively) from NE Atlantic ODP site 983 (ref. 39; 60.4° N, 23.6° W, 1,984 m water depth); percentage of warm planktic foraminifera from ODP 1063; synthetic Greenland temperature record[13]; calculated gross freshwater flux due to melting continental ice sheets[25]. **(b)** (from top to bottom) εNd from ODP 1063 (NW Atlantic, purple symbols this study, grey symbols from ref. 17); SS mean grain size from ODP 983 (NE Atlantic, error bars are 1σ); benthic foraminiferal δ$^{13}$C; benthic foraminiferal δ$^{18}$O; reconstructed sea level[3]. Pink box represents inferred overshoot of AMOC during early MIS 5e, blue boxes represent the period of weakened circulation during HS11.

retreat of the North American ice sheet increased the weathering and supply of old continental material (with very negative $\varepsilon_{Nd}$) to a broader area of the surface NW Atlantic. Irrespective of the exact location of convection though our results suggest that AMOC recovery following HS11 was accomplished (at least in part) by a drastic deepening of deep waters formed in the NW Atlantic. Similarly negative $\varepsilon$Nd values have been documented at the location of ODP site 1063 during the early Holocene[21,35]. In particular, Howe *et al.*[35] conclude that these very negative early Holocene $\varepsilon$Nd values might reflect the 're-labelling' of deep waters in the Labrador Sea by interaction with particularly un-radiogenic sediments, followed by their southward advection

to abyssal depths. A key question remains as to whether these deep waters originated in the NW or NE Atlantic.

Results from an Earth System Model experiment (run in coarse resolution)[36] suggest that convection in the South Labrador Sea could reach depths of 3,000–4,000 m during an abrupt deepening (and overshoot) of the AMOC following a period of weakened overturning (at least under glacial boundary conditions). Results of that experiment suggest that an overshoot of the AMOC occurs due to the accumulation of heat and salt in the intermediate depth tropical Atlantic, which enters the South Labrador Sea and induces hydrostatic instabilities. Such an increase in salinity would result in greater densities that could help displace colder

southern-sourced deep waters and reinvigorate the northern cell of the AMOC. In this respect the initiation of deep water convection in the NW Atlantic could effectively act as the trigger for AMOC resumption more broadly. However, we note that the simulated overshoot is only a transient (decadal) feature of the model simulation whereas the negative spike we observe in $\varepsilon_{Nd}$ lasts for thousands of years.

Our results require that deep waters formed in the NW Atlantic were dense enough (at least relative to other deep waters present at that time) to influence the deepest parts of the Atlantic basin throughout the earliest part of MIS 5e. This would be possible only if other sources of deep water (which today represent the densest waters within the Atlantic) were diminished or possessed lower densities than those forming in the NW Atlantic. Indeed, a partial solution to this conundrum is hinted at by a recent proxy reconstruction from the Southern Ocean[37], which suggests that the formation and or density of AABW around Antarctica was greatly reduced across the same interval as we invoke enhanced deep-water formation in the NW Atlantic.

Furthermore, evidence from the deep NE Atlantic suggests that deep water formation in the Nordic Seas and its overflow into the Atlantic as L-NADW may not have resumed its typical interglacial mode until ~124 ka (ref. 18). Our new SS measurements from ODP Site 983 (situated on the Gardar Drift, southeast of Iceland) provide further insight into this possibility (Fig. 3). SS mean grain size of the terrigenous fraction of marine sediments can be used as a proxy for the flow speed of bottom currents[38]. The site of ODP Site 983 is sensitive to the overflow of dense waters formed in the Nordic Seas across the Iceland-Scotland Ridge (so-called Iceland-Scotland Overflow Water (ISOW)), an important pre-cursor to L-NADW. Our new record of SS shows a strong increase ~124 ka, much later than our inference of AMOC recovery via deep water formation in the NW Atlantic ~129 ka. While the record of SS from a single water depth cannot tell us about the gross flux of ISOW it is nevertheless instructive. For example, a depth transect of SS records from the same region (including site 983) covering the Holocene[24] reveals a gradual deepening and strengthening of ISOW over the course of the early Holocene, with maximum inferred flow speeds (highest SS) at the site of ODP Site 983 being attained ~7 ka, when the ISOW was inferred to have reached its present-day depth and maximum net strength. By analogy (and acknowledging the limitations of a single core site) we infer from our record that ISOW strengthened, and or deepened (becoming denser with respect to surrounding water masses) ~124 ka.

In Fig. 3a we show surface records from ODP Sites 983 (ref. 39) and 1063 (this study). Both records suggest an abrupt warming of the surface ocean ~129 ka but while site 1063 experienced its warmest temperatures during early MIS 5e, the record from site 983 suggests that optimum conditions were not attained until ~124 ka towards the northeast. A similar finding was reported previously[40,41] and interpreted as the delayed recovery of a full interglacial mode of circulation, with reduced inflow of warm waters to the Nordic Seas via the North Atlantic Current during early MIS 5e. The sustained occurrence of ice rafting in the high latitude North Atlantic until ~124 ka is evidenced by the record of IRD from ODP Site 983 (ref. 39; Fig. 3a) as well as previous studies in the Nordic Seas[41]. Correspondingly fresher conditions across the Nordic Seas and NE Atlantic could explain the decrease in formation and or density of ISOW during early MIS 5e (ref. 18) and the resultant density 'vacuum'[42] that may have allowed NW Atlantic deep waters to reach abyssal depths.

**Rapidity of deep ocean change**. Thanks to its high temporal resolution our new record of benthic foraminiferal $\delta^{18}O$ provides further evidence for the timing and rapidity of ocean circulation

change at the onset of MIS 5e in the NW Atlantic (Figs 2 and 3). The record reveals a very large (0.90 ± 0.14‰) and abrupt decrease in $\delta^{18}O$ at the same time as we observe surface ocean warming at the end of HS11. The transition takes place in <400 year (occurring between two samples, Methods section) and occurs after the main phase of deglacial sea-level rise[3] (Fig. 3b), hence it cannot be explained simply by a whole-ocean change in $\delta^{18}O$. More likely it reflects a change in water mass geometry and the relative dominance of water masses with very different temperature/salinity characteristics within the abyssal North Atlantic. Equally rapid changes in deep ocean circulation in the same area across MIS 5e/d were reported previously[43].

Changes in benthic foraminiferal $\delta^{18}O$ can reflect changes both in bottom water temperature and the oxygen isotopic composition of seawater ($\delta^{18}O_{sw}$ or $\delta w$), which is related to salinity[44]. The most recent calibration for the temperature sensitivity of cosmopolitan benthic foraminifera is $-0.25$‰ per °C for cold waters[45] so a shift of $-0.90$‰ in benthic foraminiferal $\delta^{18}O$ implies a warming of 3.6 °C given no change in $\delta w$. Modern deep waters formed in the Southern Ocean are fresher and have lower $\delta w$ than more northerly intermediate waters (the modern offset in $\delta w$ between AABW and U-NADW is ~0.5‰ (ref. 44)). If the observed shift in benthic foraminiferal $\delta^{18}O$ ~129 ka reflected a change from an equivalent of modern AABW to modern U-NADW the net shift of $-0.90$‰ would require a warming of 5.6 °C, which is similar to the modern temperature difference between AABW and LSW. On the other hand, pore water studies suggest that glacial-age (MIS 2) southern deep waters may have been significantly more saline (with higher $\delta w$ by 0.12–0.42‰) than northern water masses[46]. It is not possible to know at this stage whether such values would be applicable to the transition from MIS 6 to MIS 5e, but if they were then a shift from southern to northern deep water masses would require a temperature increase of 1.9–3.1 °C to produce a net change of $-0.90$‰ in benthic $\delta^{18}O$. Furthermore, since the deglacial rise in sea level across T2 was only just complete by 129 ka (refs 3,25; Fig. 3b) it is entirely feasible that the corresponding change in $\delta w$ had not fully penetrated to all parts of the ocean interior[47]. If the deglacial evolution of $\delta w$ in southern-sourced deep waters lagged behind that of northern sources this could also have contributed to the sharp decrease in benthic $\delta^{18}O$ we observe ~129 ka as the influence of southern waters gave way to those originating from the north.

Another possible mechanism that could explain the shift towards lighter benthic $\delta^{18}O$, as well as the very negative $\varepsilon Nd$ values ~129 ka without invoking subsidence of a 'low density' water mass, is the formation of dense brines possibly through wind action over coastal polynyas in Baffin Bay (analogous to those formed around modern-day Antarctica and within the Arctic Ocean[48]), perhaps as a result of anomalous wind patterns during the earliest part of MIS 5e. However a number of studies suggest that dissolved inorganic carbon is preferentially rejected relative to alkalinity during brine formation, possibly as a result of $CaCO_3$ precipitation and subsequent entrapment within the sea ice matrix while aqueous $CO_2$ escapes e.g. ref. 49. This would have the effect of decreasing the carbonate saturation state of deep waters formed in this way and consequently we might not expect to observe enhanced preservation at the site of ODP Site 1063 during early MIS 5e (Fig. 2b) if deep waters at the site were being formed through brine rejection. Notwithstanding, the possibility that brine formation may have contributed to the negative $\varepsilon Nd$ values we observe deserves further consideration.

**Deglacial rise in atmospheric $CO_2$**. Changes in ocean circulation can influence atmospheric $CO_2$ in multiple ways[8–11]. The records of atmospheric $CO_2$ and ocean circulation (as inferred from

North Atlantic Nd isotopes) across the last two glacial terminations are shown in Fig. 4. The contrast between glacial (low $CO_2$) and interglacial (high $CO_2$) conditions has led to a plethora of hypotheses as to the mechanisms controlling atmospheric $CO_2$ on orbital timescales (see ref. 8 for a summary) with an overall consensus that changes in the oceanic storage of carbon (through synergistic interactions between physical, chemical and biological processes) are the most important. But of relevance to this study are the transitions themselves between glacial and interglacial state, which appear to proceed through mechanisms operating on sub-millennial to millennial timescales.

Atmospheric $CO_2$ increased over several discrete intervals across T1 (ref. 5; Fig. 4). During times of weakened and/or shallow AMOC (HS1 and the YD) $CO_2$ increased relatively gradually ($\sim 10$ p.p.m.v. kyr$^{-1}$). This may be contrasted with two distinctly more abrupt increases ($\sim 10$–$15$ p.p.m.v. in $100$–$200$ years) that occurred on recovery to a stronger mode of AMOC following HS1 and the YD[50]. In fact the cycle of gradually rising $CO_2$ during times of particularly weak or shallow AMOC (HS events), followed by an abrupt increase on recovery is not unique to glacial terminations and is observed repeatedly throughout the last glacial period. For example, the transition from HS4 into Dansgaard-Oeschger (D-O) event 8 was marked by an abrupt increase in atmospheric $CO_2$ of $\sim 10$ p.p.m.v. (ref. 51) as the AMOC deepened[20]. A similar pattern marked the end of HS5 and the onset of D-O events 19–21 (refs 17,52,53). Previous studies

suggest that enhanced vertical mixing within the Southern Ocean during times of reduced AMOC[54,55], combined with a replacement of NADW by AABW (which has a higher preformed nutrient content) could promote the gradual rise in $CO_2$ at these times[11,56]. In addition, a reduction in northern hemisphere land vegetation due to a southward shift of the Intertropical Convergence Zone could contribute to $CO_2$ rise during times of weakened AMOC[10].

The much more rapid increases in atmospheric $CO_2$ following HS1 and the YD (and presumably equivalent events during MIS 3) are thought to be linked to resumption of the AMOC[5,50], possibly a result of the fast changes in solubility (as a function of temperature and salinity) associated with AMOC recovery[9] and the flushing of respired carbon from the deep ocean as the AMOC deepens[20]. Rapid thawing of boreal permafrost and increased respiration of soil-bound carbon stocks could have provided an additional source of carbon at these times[57]. The subsequent and more gradual decrease in $CO_2$ observed while the AMOC is in a strong mode is thought to reflect the reversal of processes driving its increase during intervals of weakened circulation[11]. Thus the abrupt rise in atmospheric $CO_2$ associated with a strengthening of AMOC is only a transient feature, reflecting the different timescales of the mechanisms involved (for example, the rapid effects of decreased solubility[9] and deep ocean flushing[20] driving up $CO_2$, in contrast to the subsequent build-up of regenerated carbon in the deep ocean[11] driving $CO_2$ back down).

The close relationship between atmospheric $CO_2$ and the AMOC described above suggests that when ocean circulation is in quasi equilibrium (which arguably is the case only during full interglacial and full glacial conditions[13,58]) then $CO_2$ should remain (approximately) constant. Of course, additional drivers such as carbonate compensation (for example, during the Holocene[59]) and fossil fuel burning (e.g. within the Anthropocene) may affect $CO_2$ independently of ocean circulation on a variety of timescales.

Building on these arguments we now compare the last two terminations (Fig. 4). Atmospheric $CO_2$ increased during the intervals of weak/shallow AMOC associated with HS1 and the YD (T1) and HS11 (T2). Following the continuous rise in $CO_2$ throughout HS11 a transient maximum was attained on recovery and (inferred) overshoot of the AMOC during early MIS 5e, after which $CO_2$ stabilized at an interglacial level as the AMOC resumed its interglacial mode by $\sim 124$ ka. In contrast, the transient maximum in atmospheric $CO_2$ associated with the B-A occurred within its overall deglacial rise across T1 and its effect is therefore obscured (essentially discounted) within the net change in $CO_2$. Moreover, the abrupt rise in $CO_2$ associated with AMOC recovery following the YD was apparently smaller than that following HS11 (perhaps reflecting the shorter duration of the YD) and in combination with the very long duration of HS11 the overall change in $CO_2$ across T2 was larger (by $\sim 20$ p.p.m.v.) than that across T1. On the other hand, allowing for the transient maxima in $CO_2$ associated with AMOC recovery, the net change in $CO_2$ between glacial and interglacial conditions was actually quite similar across both terminations (Fig. 4). We therefore suggest that the apparently larger increase in $CO_2$ across T2, as compared with T1 was a result of the long-lasting AMOC perturbation associated with HS11 and consequently its late resumption at the beginning of MIS 5e. Note that we do not know how or if the location of AMOC resumption (NW versus NE Atlantic) may affect the magnitude of the transient maximum in $CO_2$. Our arguments here revolve around the timing of recovery only. We also acknowledge that the majority of our discussion is based on findings from a single core site in the NW Atlantic. Future validation of our results will require equivalent

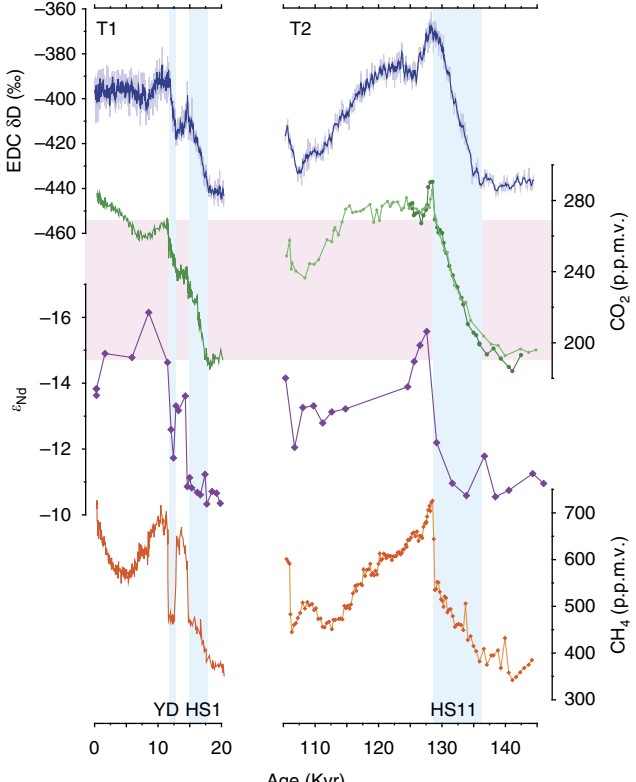

**Figure 4 | Evolution of ocean circulation and $CO_2$ across two terminations.** Records of (from top to bottom) Antarctic temperature ($\delta D$ proxy)[1], atmospheric $CO_2$ (refs 2,5,6), seawater $\varepsilon_{Nd}$ (ref. 21 and this study) and atmospheric $CH_4$ (ref. 69). Blue boxes are intervals of weakened and/or shallow AMOC; pink box encompasses the deglacial change in $CO_2$ across T1. YD, HS1 and HS11 are Heinrich stadials. $CO_2$ record across T1 is from ref. 5 on the WDC06A-7 timescale. $CO_2$ records across T2 are from ref. 6 (dark green) and ref. 2 (light green), both on the AICC2012 timescale[23]. $CH_4$ records are both on the AICC2012 timescale.

reconstructions from a variety of sites across a much broader region.

A final question concerns why there was no YD-like event during T2. It is thought that recovery of the AMOC during deglaciation may occur with the cessation of freshwater release across the North Atlantic[31] or in response to more gradual global warming, in which case the addition of freshwater may still act to delay resumption[32]. The lack of an early recovery during T2 could therefore reflect the larger insolation forcing and faster ice sheet retreat associated with the penultimate termination[33], providing a sustained supply of freshwater to regions of deep water formation and delaying resumption of the AMOC until atmospheric $CO_2$ had reached its interglacial level.

## Methods

**Sample preparation.** ODP Site 1063 core was resampled every 4 cm along the shipboard splice across the interval of interest. Sediment core samples were washed and sieved at 63 μm before drying and weighing. Planktonic foraminiferal species and fragment counts were performed on splits of the >150 μm fraction containing ~300 individual tests. Per cent fragmentation is calculated following Le and Shackleton[60]. Stable isotopes were measured on the planktic species *G. inflata* picked from the 300 to 355 μm fraction. Due to very low abundances of benthic foraminifera, 3 species (*C. wuellerstorfi*, *Melonis pompilioides* and *Oridorsalis umbonatus*, all picked from >150 μm and analysed individually) were used to obtain a more complete record. Measurements were performed at Cardiff University stable isotope facility using a ThermoFinnigan MAT-252 mass spectrometer (long-term external reproducibility better than ± 0.08‰ for δ[18]O and ± 0.03‰ for δ[13]C) for benthic samples and a Delta Advantage V (long-term external reproducibility ± 0.1‰ for δ[18]O) for planktics. Offsets in δ[18]O between benthic species were accounted for by correcting to *C. wuellerstorfi* by subtracting the average offset between species as measured in samples where multiple species were present (*M. pompilioides* − 0.15‰, *O. umbonatus* − 0.38‰). All results are reported within Supplementary Data 1.

**Fossil fish teeth and debris.** Fossil fish teeth and debris were handpicked from the >63 μm sediment fraction of 91 samples at ODP Site 1063 between 34.0 and 39.9 metres composite depth (mcd). To obtain enough material for Nd isotope analyses, up to five samples were combined as indicated in Supplementary Data 1. The teeth and debris were cleaned with ultrapure Milli-Q water (18.2 MΩ water) and methanol (that is, no reductive and oxidative cleaning), following[61]. Samples were digested in 2M HCl, dried down, converted to nitrate from and subjected to a standard two-stage ion chromatography procedure in the MAGIC clean room laboratories at Imperial College London. In brief, Eichrom TRU-Spec resin (100–120 μm bead size) was utilized to isolate the REEs from the sample matrix and Eichrom LN-Spec resin (50–100 μm bead size) was utilized to separate Nd from the other REEs (slightly modified after ref. 62).

**Neodymium isotope ratios.** Neodymium isotope ratios were measured on a Nu Plasma HR MC-ICP-MS and a Thermo Scientific Triton TIMS at the MAGIC Laboratories at Imperial College London. Measurements on the MC-ICP-MS were carried out in static mode, using a $^{146}Nd/^{144}Nd$ ratio of 0.7219 to correct for instrumental mass bias following the exponential law. $^{144}Sm$ interferences can be adequately corrected if the $^{144}Sm$ contribution is <0.1% of the $^{144}Nd$ signal, which was the case for all samples. Measured $^{143}Nd/^{144}Nd$ ratios of the JNd$_i$ standard yielded ratios of 0.512133 ± 0.000013 (2SD, $n = 8$) and 0.512056 ± 0.000015 (2SD, $n = 27$) during two separate sessions. Measurements on the Thermal Ionisation Mass Spectrometer (TIMS) were carried out as Nd oxides (NdO$^+$) following the method outlined by Crocket *et al.*[63], yielding JNd$_i$ $^{143}Nd/^{144}Nd$ ratios of 0.512101 ± 0.000007 (2SD; $n = 5$). Accuracy was achieved by correcting all sample results from both machines to the published JNd$_i$ $^{143}Nd/^{144}Nd$ ratio of 0.512115 ± 0.000007 (ref. 64), and confirmed with USGS rock standard BCR-2 results on both machines, which were within error of the recommended value by Weis *et al.*[65] Comparability between both machines was furthermore demonstrated by excellent agreement of duplicate measurements for four samples (Supplementary Data 1). Procedural blanks were consistently below 10 pg Nd.

The data used to create the (sub)surface map of seawater Nd isotopic compositions in the North Atlantic (Fig. 1a) were assembled from the compilation by van de Flierdt *et al.*[27] For each available station, Nd isotope results for the uppermost water depth were utilized if this depth was <65 m. One exception was made in the Labrador Sea, where a measurement from 100 m depth was included. For the Baffin Bay area north of the shallow sill separating it from the Labrador Sea, data from the entire water column were integrated. All stations utilized for the compilation are indicated by small black dots on the (sub)surface ε$_{Nd}$ map. Station locations from the northern part of the GEOTRACES transect GA02, sampled for dissolved Nd isotopes, are indicated by black circles on the small map (Fig. 1b).

Water depths for all Nd samples are indicated by small black dots on the section (Fig. 1c).

**Age model development.** Since we wish to compare our records directly with those from ice cores we need to refine earlier versions of the age model for ODP Site 1063 across T2 that were based on orbital and paleomagnetic approaches[66,67]. In a recent study[52] we derived an age model for the same core across the MIS 5a/4 boundary by tuning between a high resolution record of planktic δ[18]O (measured on *G. inflata*) and the Greenland ice core temperature record. Abrupt shifts in planktic δ[18]O (including for *G. inflata*) in the Northwest Atlantic are thought to have been synchronous to the shifts in Greenland ice core δ[18]O across Termination 1 and throughout MIS 3 (refs 22,68). Although planktic δ[18]O from the subtropical Northwest Atlantic during D-O events likely contains both temperature and salinity signals, the 'raw' planktic δ[18]O appears in-phase with Greenland climate, at least on multi-centennial and longer timescales[52]. Moreover, our planktic foraminifer species count records share many similarities with the isotope record (Fig. 2) and we use these to support our tuning strategy. Because the Greenland record does not encompass Termination 2 we instead use the record of atmospheric methane from Antarctica[69] on the AICC2012 timescale[23] as a tuning target, supplemented by the synthetic record of Greenland temperature variability, GL$_T$_syn[13]. Sharp increases in $CH_4$ are consistently aligned (within ~60 year) with rapid shifts in Greenland temperature during the last 120 kyr (ref. 70).

Age uncertainties in our approach derive from the precision of alignment between the various records and the absolute uncertainty of the ice core age model. Because we are here interested in the relative timing of marine events with respect to the ice core record our error analysis does not consider the additional uncertainty of the ice core chronology. In the case of the implied resumption of deep overturning circulation following HS11, we note that the − 0.9‰ shift in benthic δ[18]O occurs across the same interval (between two samples, that is, within 400 year or within 300 year if our tie-point is placed at the end of the transition instead of midway through) as the warming implied by our planktic δ[18]O and faunal records at the end of the HS11, which we tie to the abrupt rise in $CH_4$ at ~128.7 ka (Fig. 2). The abrupt increases in both $CH_4$ and $CO_2$ at this time occurred in parallel between 128.9 and 128.5 ka on the AAIC2012 age model[2,23]. Therefore because we make the assumption of synchronicity between surface ocean temperature variability and northern hemisphere climate, as reflected by $CH_4$ and GL$_T$_syn, we estimate that that the recovery of deep overturning circulation within the North Atlantic was synchronous with the abrupt rise in $CO_2$ to within 400 year (the width of the transitions in $CO_2$, $CH_4$ and benthic δ[18]O).

**Sortable silt measurements.** Samples from ODP Site 983 were prepared for SS analysis following established protocols[24,38]. Briefly, 2–4 g of bulk fine fraction (<63 μm) was treated with acetic acid and sodium carbonate to remove carbonate and biogenic silica, respectively. The residual silicate fraction was treated with Calgon and ultrasonicted for 4 min before analysis on a Beckman Coulter Multisizer 3 coulter counter. At least two replicate measurements of the arithmetic mean calculated from the differential volume of grains within the 10–63 μm terrigenous silt fraction are reported for each sample depth. The average s.d. between replicate measurements for all samples is ± 0.23 μm.

**Data Availability.** All data generated or analysed during this study are included in this published article (and its Supplementary Information files).

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

## Acknowledgements

We thank Katharina Kreissig, Sian Lordsmith, Stephen Conn and Lindsey Owen for laboratory assistance, Torben Struve for help with Nd isotope analyses, and David Thornalley, Sophie Nuber and Gregor Knorr for discussions. The SS data from ODP 983 were produced by Fiona Piggott (Cardiff University) as part of her MESci undergraduate thesis project in 2012/13. This research used samples provided by the Integrated Ocean Drilling Programme (IODP). We acknowledge support from UK NERC (grants NE/J008133/1, NE/J021636/1, and NE/L006405/1) and a President's Research Scholarship at Cardiff University (ED).

## Author contributions

E.L.D. performed laboratory work on material from ODP 1063 with assistance from those mentioned in the acknowledgements. All authors helped design the project, interpret datasets and write the paper.

## Additional information

**Competing financial interests:** The authors declare no competing financial interests.

**How to cite this article**: Deaney, E. L. *et al.* Timing and nature of AMOC recovery across Termination 2 and magnitude of deglacial $CO_2$ change. *Nat. Commun.* **8**, 14595 doi: 10.1038/ncomms14595 (2017).

**Publisher's note**: 

