## [Peer Review File · Nature Communications]

Reviewers' comments:

Reviewer #1 (Remarks to the Author):

Deaney et al. present a new combined data set of stable isotopes and neodymium isotopes (eNd) from the NW Atlantic (ODP1063) mainly covering the time range of T2. Specifically, the authors have successfully extracted the authigenic Nd isotopic composition from fossil fish debris. This is hard won data for which the authors deserve credit. These new measurement results are compared with existing data sets: cores of the northern North Atlantic (ODP983, ODP984), ice-core records and in particular with published data sets from ODP1063. The main conclusion of this study is to link the variability (amplitudes) of CO₂ changes across T1 and T2 with changes in ocean circulation

Here the authors pursue an important issue of paleoclimatology. They do so by following a rational, promising and timely analytical approach. They provide a large number of suggestions for explaining the observed patterns. However, I feel that there was little I have learnt from the manuscript. While the authors did a brilliant job in connecting various observations and giving a comprehensive overview on the topic they miss providing mechanistic explanations. The high number of suggestions also did not lead to a tangible conclusion. As can be seen in the case of the very last sentence the reader is still left alone with the question what was the cause for the 20 ppmv overshoot in CO₂. Was it the atmosphere or the ocean? If it was the timing of these events (what events by the way?), what is the causal chain behind it?

I would love to see this paper published, because the authors come up with a remarkable combination of proxies and ideas. But I feel they need to focus better on what they conclusively can say from the observations and how to present it. I observe a crucial lack of structure in the text (for example):

in my copy of the text the only header is "Results and Discussion" in the middle of the text, while there are no Introduction and no Conclusion;

in the introduction they need to take more care on recapping what has been published on T2. In particular I miss a detailed discussion on the differences between T1 and T2 (e.g. Carlson, A., 2008. Why there was not a Younger Dryas-like event during the Penultimate Deglaciation. *Quaternary Science Reviews* 27, 882-887).

in particular Results should be separated from Discussion. It is not obvious which data sets are new, which are published;

in the discussion they should focus on points which lead to their conclusions. There is a lot of discussion of side aspects and no clear line. E.g. there is a whole paragraph discussing the very negative eNd values. The authors found the explanation from another study too simple. While I was expecting now a quantitative or at least a more substantiated explanation they only provide further alternatives and speculations;

there is a grammar and/or content problem in the most important part of the abstract (line 21); the figures needs to be re-structured. I don't want to go too much into detail, but there is plenty of redundancy, lack of consistency (e.g. kyr or Kyr; panels a+b in Fig.1, but not in Fig. 3-6). There are too many unnecessary panels, showing the same data, while the (in my opinion) most important and very beautiful Fig.6 comes last. Captions are not sufficient sometimes (e.g. Fig. 5). To increase readability panels should be labeled with the location/archive of the shown data. At least once error bars needs to be shown.

I welcome very much the authors presenting the Nd isotopic data in table 1. But I think they should also give the stable isotopes data.

As far as I understood this study is mainly a Nd-based revisit of ODP1063 following up Bohm et al. 2015. Here >20 new Nd data points are presented. Following Fig. 4, these are less or equal the number of data points provided by Bohm et al. during the time period of interest. But the authors claim that they provide a better age model. In Fig.4 the eNd data sets from both studies are shown with different age models. But if the main novelty of this study is a new age model for T2, then the authors need to merge their data with the Bohm et al. results in order to provide a higher time resolution. If they are confident with the new age model they should also give a table with the Bohm data based on this new age model. Bohm et al. did report an alternative circulation

proxy (Proactinium-Thorium ratio), which should be considered here as well, when discussing ocean circulation. Or are there well-grounded reasons why the Proactinium-Thorium proxy from the very same location and time range has not been considered? What is the purpose of showing the Bohm data from the same core with a different age model? The time resolution is not sufficient to derive a 2 kyr offset between both age models (line 123). It is 2 ka at max. But again, the data needs to be shown on a common age scale. It is not of special interest for the reader if the previous age model of Bohm et al. was wrong.

The main figure of the manuscript is clearly Fig.6. However, from the eight shown curves only one is new while reproducing a previous data set. I found a similar plot in Bohm et al. In my opinion already the conclusions on ocean circulation by Bohm et al. need to be used with care given that a single core location in the Atlantic is considered to be exemplary for the whole AMOC. This general problem did not change with the here presented alternative data subset. For me this raises the question on the novelty of the story, given the relatively vague conclusions on the reasons for the differences between T1 and T2 (e.g. last paragraph). What is the mechanism behind a "late recovery" of AMOC leading to an excess of 20 ppmv CO₂? What are the differences in the AMOC overshoots of T2 to T2? There is an overshoot implied by the data by Roberts et al. 2010. Synchronous Deglacial Overturning and Water Mass Source Changes. *Science* 327, 75-78, and suggested before by other studies: e.g. Marson et al. 2015. Evolution of the deep Atlantic water masses since the last glacial maximum based on a transient run of NCAR-CCSM3. *Climate Dynamics*, 1-13; Thornalley et al. 2013. Long-term variations in Iceland-Scotland overflow strength during the Holocene. *Climate of the Past* 9, 2073-2084.

Reviewer #2 (Remarks to the Author):

The role of the Atlantic's overturning circulation in Earth's climate change, especially during the termination of the Pleistocene ice ages, is a topic of widespread interest. Consequently, the paper by Deaney et al., showing an abrupt warming and deepening of Atlantic overturning circulation at the end of Termination II, represents a welcome contribution.

The high resolution faunal assemblages nicely resolve the climate oscillations over the time interval investigated so they will be useful for future studies as well as for the interpretation presented here. One would like to have seen Nd isotope data at higher resolution, but the existing record leaves little doubt about the overshoot at the end of Termination I. The benthic foraminifera carbon isotope record is most difficult to interpret in terms of actual changes in deep water chemistry. The wild sample-to-sample fluctuations, often in excess of one per mill, must reflect some kind of unidentified artifact. It would be good if the authors offered their thoughts about the cause of this variability. Despite the unrealistic variability, the smoothed benthic $\delta^{13}C$ record is consistent with the overall interpretation.

The remaining comments are divided by category.

INTERPRETATION

1) The principal weakness of the paper is that the reasoning that leads from the data presented by the authors to the main conclusion expressed at the end of the abstract, "Association between a late resumption of AMOC and the deglacial CO₂ overshoot ~129ka suggests that had circulation recovered earlier during T2 then the net change in CO₂ across the penultimate deglaciation may have been smaller and more similar to that which occurred across T1," is never developed very well.

What does this imply in terms of ocean processes? Is the overshoot a transient feature that will dissipate with time without any external forcing? How is the overshoot at the end of termination II related to the rapid but smaller rises in atmospheric CO₂ at the end of Heinrich Stadial (HS) I and at the end of the Younger Dryas? If the net glacial-to-interglacial change in ocean water mass

structure and chemical composition is essentially the same across Terminations II and I, then how does the precise sequence of events during the termination alter the ultimate level of CO₂ reached in the atmosphere. This is not to argue that the authors are incorrect in their interpretation; only that their interpretation is incomplete so it is impossible to judge whether or not the interpretation is reasonable in terms of our understanding of the processes that regulate the exchange of CO₂ between the ocean and the atmosphere.

Without further explanation of the processes involved, the concluding comments (Lines 237 - 247) seem to be speculation with very slim support.

2) Related to Point 1 above, do the authors see any role for insolation in creating the different response of atmospheric CO₂ across the two terminations? The rise in northern hemisphere summer insolation was much greater across Termination II. Could this have been a factor?

3) Adkins et al. (1997) present similar records from the Bermuda Rise for core MD95-2036 so it is surprising that Deaney et al. do not compare their results to those of Adkins et al. Such a comparison is necessary because one sees no short-lived overshoot at the end of Termination II in the benthic d₁₈O record of Adkins et al., where the benthic d₁₈O values are quite stable from approx. 128 to 118 ka.

If one takes the liberty of adjusting the age model of Deaney et al. for ODP1063, extending the interval of ODP1063 over which benthic d₁₈O values (Figure 1b bottom) fall below 3 per mill from 128 ka to 118 ka, to match the record of Adkins et al., then two things happen. First, the benthic foraminifera d₁₃C record of Deaney et al. becomes more similar to the benthic d₁₃C record of Adkins et al., and second, the divergence of the benthic d₁₃C and d₁₈O records of ODP1063 from the records of ODP984 and ODP983 (Figure 3, bottom 2 panels) largely disappears.

Can the age models be adjusted so as to reconcile the records of Deaney and of Adkins? If so, then how does this alter the overall interpretation?

4) ODP sites 983 and 984 from intermediate depths are located on the eastern flank of the Reykjanes Ridge. Does the presence of the ridge restrict the lateral exchange of Labrador Sea water? Is it reasonable to expect vertical homogeneity of the composition of intermediate and deep water between the eastern and western basins of the North Atlantic if the ridge provides a barrier to lateral exchange?

In contrast to the intermediate-depth sites in the eastern basin (ODP 983 and 984), Hodell et al. (2009) and Galaasen et al. (2014) documented that benthic d₁₃C at truly deep sites reached maximum interglacial values significantly later (well into the MIS5e interglacial) than reported here in the new data from the authors. Could this be simply an age model problem?

PRESENTATION

1) Readers will be better able to understand the data (Figure 1) if the context (Figure 2) were presented first.

2) As a general comment, readers who are not specialists in the application of the proxies exploited here will benefit if the authors provide a little more explanation of the proxies in the main text. Similarly, non-specialist readers will benefit if the authors point to specific features in their data that justify the discussion and interpretation in the main text. To illustrate, a selection of text from lines 72 - 76 is annotated in { } below:

"In agreement with 19 we reconstruct significantly less negative values ($\epsilon_{\text{Nd}} > -11$) at the end of MIS 6 and during the interval equivalent to HS11 {point reader to specific figure, record, and age that supports this statement} when we also observe the coldest surface conditions {temperature

proxies are not developed very well in this paper; first explain the proxies used to infer temperature and then point to specific features in specific records that support statements like "coldest surface conditions"} at our site (Fig. 1). In combination with low benthic $\delta^{13}\text{C}$ and poor carbonate preservation during the same interval {what is the evidence for poor carbonate preservation? Here, again, explain the proxies used to infer carbonate preservation, point to specific features in specific records that support poor carbonate preservation, and explain the link to Southern Source deep water} suggests an enhanced influence of southern sourced deep waters..."

The authors are encouraged to adopt this recommendation throughout the paper, but most notably again in the penultimate paragraph beginning on line 233 where readers will be able to better understand the statements of they are pointed to specific features in specific records.

The discussion of age control beginning on line 81 is important, but less so than explaining to readers how the various proxies are interpreted. If necessary, the information about age control can be moved to the Methods to allow more space to explain the proxies and their interpretation.

3) Lines 124 - 126: One cannot infer water mass density structure from eps-Nd! The very negative eps-Nd during early MIS5 requires a source of deep water that has no analog in the modern ocean, and that is a very interesting finding, but it provides no information about density structure of the water column.

4) Lines 151 - 154: Similar to Point 3 above, although densification of deep waters formed off northern Canada or western Greenland is not inconsistent with the eps-Nd data, the results do not require greater density of deep waters formed in this region. The depth of each water mass depends on the density of all other water masses competing for the space as well as on its own density. If freshwater released by melting of the Antarctic ice sheet caused the density of southern source deep water to have decreased during this interval, as suggested by reference 36 (line 173), then water formed in the NW Atlantic with modest density could have filled the deep basin to the bottom.

5) Paragraph beginning on Line 156: Low resolution coupled models do a poor job of simulating ocean physics, including the location and mechanism of deep water formation. Consequently, the evidence cited in this paragraph is not compelling.

6) Line 173: Inferences about water mass structure cannot be interpreted in terms of rates of water mass formation. The tendency to do this is one of the major complaints that physical oceanographers lodge against paleoceanographers. The evidence cited here suggests that the density of AABW formed during the period of interest was greatly reduced, but this cannot be equated with a greatly reduced rate of formation of AABW, or of an equivalent water mass that finds its stable position at an intermediate depth.

7) Lines 177 - 179: Here, again, the authors make inferences about absolute density whereas only relative density can be inferred. The robust conclusion from the authors' data is that the bottom water over the Bermuda Rise during the overshoot period had an eps-Nd consistent with formation in a region off NE Canada or western Greenland. The dynamically stable depth of this water mass depends as much on the density of all competing water masses as on its own density.

An alternative scenario that explains the eps-Nd and the low benthic d^{18}O (line 187) in terms of a high-density water mass on the Bermuda Rise rather than a low density water mass would be to invoke formation of cold, salty bottom water in Baffin Bay. If a reorganization of wind patterns during the overshoot interval at the end of Termination II favored coastal polynyas in Baffin Bay then salty deep waters could have formed, much as occurs today around Antarctica and in very limited regions of the Arctic Ocean. Incorporation of a relatively small amount of meltwater from summer melting of surrounding continental ice sheets could have lowered the d^{18}O of the "Baffin

Bay bottom water" so that it is not necessary to invoke unexpectedly warm temperatures to explain the light d18O values (line 198 and 203). This is not necessarily the preferred interpretation of the authors' data, but it is a plausible interpretation that cannot be ruled out. It is recommended that the authors focus on findings that are robust (e.g., eps-Nd requires a source of bottom water during the overshoot with no modern analog) and simply offer a selection of plausible explanations that can be subjects for future investigations.

SPECIFIC DETAILS:

Lines 35 - 37: The net increase in atmospheric CO₂ following Termination I was similar to that following Termination II, but the final approx. 20 ppm CO₂ rise occurred much later during the Holocene. Do the authors care to comment on this?

Line 48: Give references for the timing of Termination I.

Line 109: Good point about circulation, but then how do you explain the subsequent contrast of more than 0.5 per mil in benthic d18O?

Lines 122 - 123: Is this difference between the authors' data and the data of Böhm simply an age model difference? Are the data not from the same core? Do the records align on a depth scale? Or is this a difference in eps-Nd between fish teeth and bulk sediment leaches? The sediment leach and fossil eps-Nd in Roberts et al., (2010) look very different from each other. Could something similar affect the two records in ODP1063 across Termination II?

Lines 194 - 210: The discussion omits the most obvious influence, changes in global ice volume.

Lines 217 - 219: The inferences here are reasonable and potentially important but the statement should be supported by citing relevant work such as that of Chen et al., 2015.

ADDITIONAL COMMENTS

1) Warming in fauna and in d18O after 140 ka is not evident in the methane record, but it may be apparent in the smoothed version of the synthetic Greenland record (Barker's work) and also at nearby high northern latitude sites (Barker et al., 2015) (Mokeddem and McManus, 2016).

2) See also Lehman et al. (2002) for relevant SST data from a nearby site.

REFERENCES other than work of the authors:

Adkins, J.F., Boyle, E.A., Keigwin, L., Cortijo, E., 1997. Variability of the North Atlantic thermohaline circulation during the last interglacial period. *Nature* 390, 154-156.

Galaasen, E.V., Ninnemann, U.S., Irvani, N., Kleiven, H.F., Rosenthal, Y., Kissel, C., Hodell, D.A., 2014. Rapid Reductions in North Atlantic Deep Water During the Peak of the Last Interglacial Period. *Science* 343, 1129-1132.

Hodell, D.A., Minth, E.K., Curtis, J.H., McCave, I.N., Hall, I.R., Channell, J.E.T., Xuan, C., 2009. Surface and deep-water hydrography on Gardar Drift (Iceland Basin) during the last interglacial period. *Earth and Planetary Science Letters* 288, 10-19.

Lehman, S.J., Sachs, J.P., Crotwell, A.M., Keigwin, L.D., Boyle, E.A., 2002. Relation of subtropical Atlantic temperature, high-latitude ice rafting, deep water formation, and European climate 130,000- 60,000 years ago. *Quaternary Science Reviews* 21, 1917-1924.

Mokeddem, Z., McManus, J.F., 2016. Persistent climatic and oceanographic oscillations in the

subpolar North Atlantic during the MIS 6 glaciation and MIS 5 interglacial. *Paleoceanography*, doi: 10.1002/2015PA002813.

Roberts, N.L., Piotrowski, A.M., McManus, J.F., Keigwin, L.D., 2010. Synchronous deglacial overturning and water mass source changes. *Science* 327, 75-78.

Response to Review.

Reviewers' comments are in italics

Our responses are in standard font

Reviewer #1:

- *Deaney et al. present a new combined data set of stable isotopes and neodymium isotopes (ϵNd) from the NW Atlantic (ODP1063) mainly covering the time range of T2. Specifically, the authors have successfully extracted the authigenic Nd isotopic composition from fossil fish debris. This is hard won data for which the authors deserve credit. These new measurement results are compared with existing data sets: cores of the northern North Atlantic (ODP983, ODP984), ice-core records and in particular with published data sets from ODP1063.*

In fact all of the datasets from ODP 1063 presented here (except for the Bohm ϵNd data) were produced as part of this project and were not previously published. We obviously did a bad job at making this clear in the original submission and have reworked accordingly (see also response to later comment).

- *The main conclusion of this study is to link the variability (amplitudes) of CO₂ changes across T1 and T2 with changes in ocean circulation. Here the authors pursue an important issue of paleoclimatology. They do so by following a rational, promising and timely analytical approach. They provide a large number of suggestions for explaining the observed patterns. However, I feel that there was little I have learnt from the manuscript. While the authors did a brilliant job in connecting various observations and giving a comprehensive overview on the topic they miss providing mechanistic explanations. The high number of suggestions also did not lead to a tangible conclusion. As can be seen in the case of the very last sentence the reader is still left alone with the question what was the cause for the 20 ppmv overshoot in CO₂. Was it the atmosphere or the ocean? If it was the timing of these events (what events by the way?), what is the causal chain behind it?*

We have wholly reworked the discussion section and included new figures to make our arguments more explicit and coherent. We have also tried to be more exclusive in the possible mechanisms we invoke.

- *I would love to see this paper published, because the authors come up with a remarkable combination of proxies and ideas. But I feel they need to focus better on what they conclusively can say from the observations and how to present it. I observe a crucial lack of structure in the text (for example): in my copy of the text the only header is "Results and Discussion" in the middle of the text, while there are no Introduction and no Conclusion;*

We have completely restructured the text within the guidelines set out by the journal. We now describe the new results under separate headings in the Results section and reserve the Discussion section for the links between ocean circulation, CO₂ and deglaciation. We hope that this format improves the overall readability.

- *in the introduction they need to take more care on recapping what has been published on T2. In particular I miss a detailed discussion on the differences between T1 and T2 (e.g. Carlson, A., 2008. Why there was not a Younger Dryas-like event during the Penultimate Deglaciation. Quaternary Science Reviews 27, 882-887).*

We have now highlighted the difference between T1 and T2 in terms of ocean circulation within the introduction. We have also addressed the question of a lack of a YD event during T2 within the discussion.

- *in particular Results should be separated from Discussion. It is not obvious which data sets are new, which are published;*

We have tried to make it clearer that all the datasets presented from ODP 1063 (apart from Bohm's ϵ Nd measurements) are new and were not previously published. Furthermore we have now included an additional unpublished dataset (sortable silt measurements from ODP 983) which we feel adds to the discussion on where AMOC resumption occurred and when. As stated we have completely separated the discussion on CO₂ from the Results section.

- *in the discussion they should focus on points which lead to their conclusions.*

We have now dedicated the discussion section to the mechanisms of CO₂ release.

- *There is a lot of discussion of side aspects and no clear line. E.g. there is a whole paragraph discussing the very negative ϵ Nd values. The authors found the explanation from another study too simple. While I was expecting now a quantitative or at least a more substantiated explanation they only provide further alternatives and speculations;*

We have gone somewhat further here and highlighted the critical observation for interpreting ϵ Nd records such as ours. We note that the modern end-members are distinguished from each other not as 'north' versus 'south' but 'south and NE' versus 'NW'. i.e. the record of ϵ Nd from the site of 1063 is not so much a record of how 'deep' the AMOC is but rather whether deep waters there formed in the south (or NE Atlantic) versus the NW Atlantic. This is quite different from the interpretation of Bohm et al and is pertinent for our conclusions because we discuss the locus of deep AMOC resumption as well as the timing. Please note though that we are not arguing that location is critical for the response of CO₂. Indeed, we do not know whether deep waters formed in the NE versus NW would have different effects on CO₂. Our arguments about CO₂ are limited to the timing of AMOC recovery and the general response of CO₂ as the AMOC recovers.

- *there is a grammar and/or content problem in the most important part of the abstract (line 21);*

We have reworked the summary.

- *the figures needs to be re-structured. I don't want to go too much into detail, but there is plenty of redundance, lack of consistency (e.g. kyr or Kyr; panels a+b in Fig.1, but not in Fig. 3-6). There are too many unnecessary panels, showing the same data, while the (in my opinion) most important and very beautiful Fig.6 comes last. Captions are not sufficient sometimes (e.g. Fig. 5). To increase readability panels should be labeled with the location/archive of the shown data. At least once error bars needs to be shown.*

We have reworked the figures. We have included error bars on the individual ϵNd measurements shown in Fig. 2. We have added annotation within the figures in an attempt to explain the various datasets in terms of our interpretations.

- *I welcome very much the authors presenting the Nd isotopic data in table 1. But I think they should also give the stable isotopes data.*

We have produced a spreadsheet of all data published here to replace Table 1 (ϵNd , stable isotopes of oxygen and carbon, planktic faunal and IRD counts, SS from ODP 983). This will be submitted as soon as the ms might be accepted. We would also be willing to send a copy for review purposes if requested.

- *As far as I understood this study is mainly a Nd-based revisit of ODP1063 following up Bohm et al. 2015. Here >20 new Nd data points are presented. Following Fig. 4, these are less or equal the number of data points provided by Bohm et al. during the time period of interest. But the authors claim that they provide a better age model. In Fig.4 the ϵNd data sets from both studies are shown with different age models. But if the main novelty of this study is a new age model for T2, then the authors need to merge their data with the Bohm et al. results in order to provide a higher time resolution.*

We have now included the Bohm dataset on our revised age model within Figs. 2 and 3. We have not merged the two datasets because they were derived through very different procedures and we believe that our record (based on fish debris) gives a more faithful record of bottom water composition. It is slightly lower resolution than the Bohm dataset because fish debris is a very minor component of these sediments, requiring the combination of multiple samples for each measurement.

- *If they are confident with the new age model they should also give a table with the Bohm data based on this new age model.*

We have included the Bohm dataset on our revised age model within the spreadsheet described above.

- *Bohm et al. did report an alternative circulation proxy (Proactinium-Thorium ratio), which should be considered here as well, when discussing ocean circulation. Or are there well-grounded reasons why the Proactinium-Thorium proxy from the very same location and time range has not been considered?*

We have added reference to the Pa/Th data from the Bohm et al., study but have not included their results within a figure. While we do not disagree with the overall conclusions of the

Bohm study, we believe that the Pa/Th proxy at this site can be influenced by opal production and export and that such productivity can change in parallel with climate, leading to possible over- or mis-interpretation of Pa/Th in terms of ocean circulation (e.g. Keigwin and Boyle, 2008, *Paleoceanography* doi:10.1029/2007PA001500; Griffiths et al., 2013, *Paleoceanography*, doi:10.1002/palo.20030; Hayes et al., 2015, *Deep Sea Research II*, <http://dx.doi.org/10.1016/j.dsr2.2014.07.007>).

- *What is the purpose of showing the Bohm data from the same core with a different age model?*

Please note that we plotted the two datasets on their respective (different) age models in this manuscript in response to an earlier reviewer's comment on a previous incarnation of this paper where we were told that there was nothing novel about our result because the two datasets were in agreement (which they were only because we had plotted both on our new age model). We therefore decided to make the difference more explicit in this manuscript but agree that there is not much point in using two age models for a single core and are glad to be able to plot the two datasets on the same age model in this revised version.

- *The time resolution is not sufficient to derive a 2 kyr offset between both age models (line 123). It is 2 ka at max.*

This is correct with regards to the age model used by Bohm but we wanted to make the point that our age model is different to the original age model and that that was not precise enough to enable comparison with the ice core record of CO₂. Our new datasets have much better resolution (<300 years across T2) and we are now able to compare our records with the ice cores. The record of benthic δ¹⁸O provides the most compelling case for the abruptness of the deep water transition because it too has high resolution, much higher than either of the εNd records. We have removed the reference to 2kyr.

- *But again, the data needs to be shown on a common age scale. It is not of special interest for the reader if the previous age model of Bohm et al. was wrong.*

Done

- *The main figure of the manuscript is clearly Fig.6. However, from the eight shown curves only one is new while reproducing a previous data set. I found a similar plot in Bohm et al. In my opinion already the conclusions on ocean circulation by Bohm et al. need to be used with care given that a single core location in the Atlantic is considered to be exemplary for the whole AMOC. This general problem did not change with the here presented alternative data subset. For me this raises the question on the novelty of the story, given the relatively vague conclusions on the reasons for the differences between T1 and T2 (e.g. last paragraph). What is the mechanism behind a "late recovery" of AMOC leading to an excess of 20 ppmv CO₂? What are the differences in the AMOC overshoots of T2 to T2? There is an overshoot implied by the data by Roberts et al. 2010. Synchronous Deglacial Overturning and Water Mass Source Changes. *Science* 327, 75-78, and suggested before by other studies: e.g. Marson et al. 2015. Evolution of the deep Atlantic water masses since the last glacial maximum based on a transient run of NCAR-CCSM3. *Climate Dynamics*, 1-13;*

Thornalley et al. 2013. Long-term variations in Iceland-Scotland overflow strength during the Holocene. Climate of the Past 9, 2073-2084.

We have added significantly to the discussion on the differences between T1 and T2 and have been bolder in our conclusions about the mechanisms involved. We have also gone further in analysing the last 5 terminations (as far as is possible given no ϵNd data). We hope that our paper now provides some novelty with respect to the Bohm et al., study.

Reviewer #2:

- *The role of the Atlantic's overturning circulation in Earth's climate change, especially during the termination of the Pleistocene ice ages, is a topic of widespread interest. Consequently, the paper by Deaney et al., showing an abrupt warming and deepening of Atlantic overturning circulation at the end of Termination II, represents a welcome contribution.*

The high resolution faunal assemblages nicely resolve the climate oscillations over the time interval investigated so they will be useful for future studies as well as for the interpretation presented here. One would like to have seen Nd isotope data at higher resolution, but the existing record leaves little doubt about the overshoot at the end of Termination I. The benthic foraminifera carbon isotope record is most difficult to interpret in terms of actual changes in deep water chemistry. The wild sample-to-sample fluctuations, often in excess of one per mill, must reflect some kind of unidentified artifact. It would be good if the authors offered their thoughts about the cause of this variability. Despite the unrealistic variability, the smoothed benthic $\delta^{13}\text{C}$ record is consistent with the overall interpretation.

We have worried about the large scatter in benthic $\delta^{13}\text{C}$ and we can confirm that it is not analytical. Several other studies from this site and other sites in the same region (e.g. Keigwin et al., 1994, Nature **371** p323; Poli et al., 2000, Geology **28** p807; Poirier and Billups, 2014, Paleoceanography 10.1002/2014PA002661) also show large 'scatter' in benthic $\delta^{13}\text{C}$ even though the overall (e.g. smoothed) variability (as pointed out by the reviewer) is consistent with other proxies in terms of circulation. It is possible that the large scatter observed in benthic $\delta^{13}\text{C}$ in this region is related to changes in surface ocean productivity and export with the possibility that *C. wuellerstorfi* may occasionally become partially infaunal (and therefore reflect pore water rather than bottom water composition). However, given the coherence between the smoothed record and our other proxies we are confident that the smoothed record of benthic $\delta^{13}\text{C}$ is telling us broadly about ocean circulation.

- *The remaining comments are divided by category.*

INTERPRETATION

1) The principal weakness of the paper is that the reasoning that leads from the data presented by the authors to the main conclusion expressed at the end of the abstract, "Association between a late resumption of AMOC and the deglacial CO₂ overshoot ~129ka suggests that had circulation recovered earlier during T2 then the net change in CO₂ across the penultimate deglaciation may have been smaller and more similar

to that which occurred across T1," is never developed very well.

What does this imply in terms of ocean processes? Is the overshoot a transient feature that will dissipate with time without any external forcing? How is the overshoot at the end of termination II related to the rapid but smaller rises in atmospheric CO₂ at the end of Heinrich Stadial (HS) I and at the end of the Younger Dryas? If the net glacial-to-interglacial change in ocean water mass structure and chemical composition is essentially the same across Terminations II and I, then how does the precise sequence of events during the termination alter the ultimate level of CO₂ reached in the atmosphere. This is not to argue that the authors are incorrect in their interpretation; only that their interpretation is incomplete so it is impossible to judge whether or not the interpretation is reasonable in terms of our understanding of the processes that regulate the exchange of CO₂ between the ocean and the atmosphere.

Without further explanation of the processes involved, the concluding comments (Lines 237 - 247) seem to be speculation with very slim support

We have completely reworked and augmented our discussion on the mechanisms of CO₂ release and the connection between ocean circulation, CO₂ and deglaciation.

- *2) Related to Point 1 above, do the authors see any role for insolation in creating the different response of atmospheric CO₂ across the two terminations? The rise in northern hemisphere summer insolation was much greater across Termination II. Could this have been a factor?*

We have addressed the potential role of insolation in the discussion. We provide some support for the idea that stronger insolation could explain the lack of an early AMOC resumption during T2 by continuous and strong freshwater forcing.

- *3) Adkins et al. (1997) present similar records from the Bermuda Rise for core MD95-2036 so it is surprising that Deaney et al. do not compare their results to those of Adkins et al.*

We are grateful to the Reviewer for reminding us of this paper and we have now included reference to it. In particular that study also concluded that deep water changes could occur within 400yr.

- *Such a comparison is necessary because one sees no short-lived overshoot at the end of Termination II in the benthic d18O record of Adkins et al., where the benthic d18O values are quite stable from approx. 128 to 118 ka. If one takes the liberty of adjusting the age model of Deaney et al. for ODP1063, extending the interval of ODP1063 over which benthic d18O values (Figure 1b bottom) fall below 3 per mill from 128 ka to 118 ka, to match the record of Adkins et al., then two things happen. First, the benthic foraminifera d13C record of Deaney et al. becomes more similar to the benthic d13C record of Adkins et al., and second, the divergence of the benthic d13C and d18O records of ODP1063 from the records of ODP984 and ODP983 (Figure 3, bottom 2 panels) largely disappears. Can the age models be adjusted so as to reconcile the records of Deaney and of Adkins? If so, then how does this alter the overall interpretation?*

We have experimented with this idea and we have also looked more closely at other records from a range of depths from the North Atlantic. By doing as the Reviewer suggests we can produce a solution that is more parsimonious with a range of other records (including that of Adkins). Therefore we have adjusted the age model but we stress that our interpretation of the link between circulation and deglaciation is **not** affected by the revised age model (which only affects the interval between early and late MIS 5e). Furthermore, if one looks more closely at the age model employed for MD95-2036 it is not without its own uncertainties. For example the authors have defined tie points at the start and end of MIS 5e (which may be moveable) and they have assumed a linear relationship between focussing factor and %CaCO₃ (which presumably does not take into account changes in bottom water [CO₃²⁻]). Thus we suspect that the two cores may be more compatible than hinted at by the Reviewer. Nevertheless we are satisfied that our new age model represents a reasonable estimate over the interval of interest.

- *4) ODP sites 983 and 984 from intermediate depths are located on the eastern flank of the Reykjanes Ridge. Does the presence of the ridge restrict the lateral exchange of Labrador Sea water? Is it reasonable to expect vertical homogeneity of the composition of intermediate and deep water between the eastern and western basins of the North Atlantic if the ridge provides a barrier to lateral exchange?*

We agree that arguing for vertical homogeneity based on only 3 core sites may be viewed as naïve. Accordingly, and as a result of the Reviewer's suggestion of extending the duration of MIS 5e (see above) we have reworded the text. We note that core sites used by Curry and Oppo (Glacial water mass geometry and the distribution of $\delta^{13}\text{C}$ of ΣCO_2 in the western Atlantic Ocean 2005, *Paleoceanography*, doi:10.1029/2004PA001021) occupy positions close to the sites of ODP 983 and 984 (e.g. V29-202, V29-204, EW9302-14JPC), as well as the deep west Atlantic. Thus we state by analogue that the Atlantic shifted from a glacial-like mode to modern-like (at least with respect to $\delta^{13}\text{C}$) at 129ka and remained like this throughout most of MIS 5e. However, our ϵNd (and preservation) results suggest that conditions during the earliest part of MIS 5e were distinct from the latter part, which we ascribe to a predominance of deep water formation in the NW Atlantic as opposed to the NE. We have removed the ODP 984 dataset to avoid cluttering up the revised figures.

- *In contrast to the intermediate-depth sites in the eastern basin (ODP 983 and 984), Hodell et al. (2009) and Galaasen et al. (2014) documented that benthic $\delta^{13}\text{C}$ at truly deep sites reached maximum interglacial values significantly later (well into the MIS5e interglacial) than reported here in the new data from the authors. Could this be simply an age model problem?*

We do not think so. In fact we alluded to the findings of Hodell in the previous version and have emphasised them here with additional new sortable silt data from ODP site 983. We interpret the Hodell conclusion as representing the recovery of NE Atlantic deep water production ~124ka whereas deep water formation in the NW Atlantic recovered ~129ka.

- *PRESENTATION*

1) Readers will be better able to understand the data (Figure 1) if the context (Figure 2) were presented first.

Done

- 2) *As a general comment, readers who are not specialists in the application of the proxies exploited here will benefit if the authors provide a little more explanation of the proxies in the main text. Similarly, non-specialist readers will benefit if the authors point to specific features in their data that justify the discussion and interpretation in the main text. To illustrate, a selection of text from lines 72 - 76 is annotated in { } below:*

"In agreement with 19 we reconstruct significantly less negative values ($\epsilon_{Nd} > -11$) at the end of MIS 6 and during the interval equivalent to HS11 {point reader to specific figure, record, and age that supports this statement} when we also observe the coldest surface conditions {temperature proxies are not developed very well in this paper; first explain the proxies used to infer temperature and then point to specific features in specific records that support statements like "coldest surface conditions"} at our site (Fig. 1). In combination with low benthic $\delta^{13}C$ and poor carbonate preservation during the same interval {what is the evidence for poor carbonate preservation? Here, again, explain the proxies used to infer carbonate preservation, point to specific features in specific records that support poor carbonate preservation, and explain the link to Southern Source deep water} suggests an enhanced influence of southern sourced deep waters..."

The authors are encouraged to adopt this recommendation throughout the paper, but most notably again in the penultimate paragraph beginning on line 233 where readers will be able to better understand the statements of they are pointed to specific features in specific records.

The discussion of age control beginning on line 81 is important, but less so than explaining to readers how the various proxies are interpreted. If necessary, the information about age control can be moved to the Methods to allow more space to explain the proxies and their interpretation.

We have done as the Reviewer suggests and have given a much fuller description of all proxies and their interpretation within the Results section. We have also added annotation to the individual proxy records to help readers follow our inferences.

- 3) *Lines 124 - 126: One cannot infer water mass density structure from ϵ_{Nd} ! The very negative ϵ_{Nd} during early MIS5 requires a source of deep water that has no analog in the modern ocean, and that is a very interesting finding, but it provides no information about density structure of the water column.*

We agree (!) but we were referring to the relative densities of deep water masses. We have reworded to make this clearer. "the very negative ϵ_{Nd} values (< -15) attained during early MIS 5e demand a different deep water mass configuration relative to that of the modern North Atlantic"

- 4) *Lines 151 - 154: Similar to Point 3 above, although densification of deep waters formed off northern Canada or western Greenland is not inconsistent with the ϵ_{Nd} data, the results do not require greater density of deep waters formed in this region. The depth of each water mass depends on the density of all other water masses competing for the space as well as on its own density. If freshwater released by*

melting of the Antarctic ice sheet caused the density of southern source deep water to have decreased during this interval, as suggested by reference 36 (line 173), then water formed in the NW Atlantic with modest density could have filled the deep basin to the bottom.

Agreed and we have made it clearer that we are not necessarily arguing for a denser water mass formed in the NW Atlantic but only one that is denser than other deep waters forming in the south or NE Atlantic.

- *5) Paragraph beginning on Line 156: Low resolution coupled models do a poor job of simulating ocean physics, including the location and mechanism of deep water formation. Consequently, the evidence cited in this paragraph is not compelling.*

We acknowledge that the model used in that study has a relatively coarsely resolved ocean component but we feel that such models (so-called Earth System Models) are useful for providing ideas about questions such as that posed here. We would therefore prefer to leave this paragraph in but have tempered it with the caveat suggested by the Reviewer.

- *6) Line 173: Inferences about water mass structure cannot be interpreted in terms of rates of water mass formation. The tendency to do this is one of the major complaints that physical oceanographers lodge against paleoceanographers. The evidence cited here suggests that the density of AABW formed during the period of interest was greatly reduced, but this cannot be equated with a greatly reduced rate of formation of AABW, or of an equivalent water mass that finds its stable position at an intermediate depth.*

We agree about rates of formation versus water mass structure and we were influenced by the language of the Hayes et al., study that talks a lot about ‘reductions’, ‘shutdowns’ and ‘reinvigorations’ of deep water formation derived mainly from chemical proxies. For example the paper refers to their inferred, “...AABW shutdown during MIS 5e...”

We suggest an intermediate interpretation of their results that lies between their text and the Reviewer’s assertion: “a partial solution to this conundrum is hinted at by a recent proxy reconstruction from the Southern Ocean [Hayes et al., 2014], which suggests that the formation and or density of Antarctic Bottom Water (AABW) around Antarctica was greatly reduced across the same interval as we invoke enhanced deep-water formation in the NW Atlantic.”. We hope this satisfies the Reviewer while representing the language of the Hayes et al., study.

- *7) Lines 177 - 179: Here, again, the authors make inferences about absolute density whereas only relative density can be inferred. The robust conclusion from the authors’ data is that the bottom water over the Bermuda Rise during the overshoot period had an $\epsilon\text{-Nd}$ consistent with formation in a region off NE Canada or western Greenland. The dynamically stable depth of this water mass depends as much on the density of all competing water masses as on its own density.*

In fact we do not talk about absolute density, only relative densities. “If equivalents to the densest water masses found in the modern Atlantic (AABW and L-NADW) were not actively forming during early MIS 5e, this would provide an opportunity for potentially less-dense

deep waters formed in the NW Atlantic to maintain a presence in the abyssal North Atlantic after an initial deepening following HS11”

We have reworded this whole section to make explicit that deep water masses may have changed their relative densities as well as their rates of formation.

- *An alternative scenario that explains the eps-Nd and the low benthic d18O (line 187) in terms of a high-density water mass on the Bermuda Rise rather than a low density water mass would be to invoke formation of cold, salty bottom water in Baffin Bay. If a reorganization of wind patterns during the overshoot interval at the end of Termination II favored coastal polynyas in Baffin Bay then salty deep waters could have formed, much as occurs today around Antarctica and in very limited regions of the Arctic Ocean. Incorporation of a relatively small amount of meltwater from summer melting of surrounding continental ice sheets could have lowered the d18O of the "Baffin Bay bottom water" so that it is not necessary to invoke unexpectedly warm temperatures to explain the light d18O values (line 198 and 203). This is not necessarily the preferred interpretation of the authors' data, but it is a plausible interpretation that cannot be ruled out. It is recommended that the authors focus on findings that are robust (e.g., eps-Nd requires a source of bottom water during the overshoot with no modern analog) and simply offer a selection of plausible explanations that can be subjects for future investigations.*

Good point and we have included a discussion on this possibility. While we have found evidence that would suggest lower carbonate saturation of brines formed in this way (which would conflict with our observation of enhanced CaCO₃ preservation during early MIS 5e) we admit that further studies are needed to address this question.

- *SPECIFIC DETAILS:*

Lines 35 - 37: The net increase in atmospheric CO₂ following Termination I was similar to that following Termination II, but the final approx. 20 ppm CO₂ rise occurred much later during the Holocene. Do the authors care to comment on this?

The late Holocene rise of 20ppmv occurred on a longer timescale and is thought to represent more gradual changes (e.g. carbonate compensation). We have referred to this in passing within the revised text.

- *Line 48: Give references for the timing of Termination I.*

Done.

- *Line 109: Good point about circulation, but then how do you explain the subsequent contrast of more than 0.5 per mil in benthic d18O?*

The contrast in benthic $\delta^{18}\text{O}$ is now significantly reduced with the revised age model. However, the anomalously negative ϵNd and peak in preservation still marks early MIS5e as distinct from the latter part.

- *Lines 122 - 123: Is this difference between the authors' data and the data of Böhm simply an age model difference? Are the data not from the same core? Do the records*

align on a depth scale? Or is this a difference in eps-Nd between fish teeth and bulk sediment leaches? The sediment leach and fossil eps-Nd in Roberts et al., (2010) look very different from each other. Could something similar affect the two records in ODP1063 across Termination II?

Yes, the difference was mainly due to the age model (as discussed in response to the first Reviewer above). We have now plotted both datasets on the same age model but we have not combined the two as we feel that our dataset (derived from fish debris) should be considered separately from the bulk leachate record of Bohm et al.

- *Lines 194 - 210: The discussion omits the most obvious influence, changes in global ice volume.*

Yes, although the change in ice volume across T2 was essentially complete by 129ka (according to recent work by Grant, Rohling and others and as mentioned in the previous version) there is a good bet that the full change in δw had not penetrated to all parts of the ocean. Moreover it is likely that the deep waters influencing the site of 1063 during HS11 saw the signal (that of decreasing δw) later than those northern-sourced waters that replaced them ~129ka and this could therefore explain a part of the abrupt decrease in $\delta^{18}\text{O}$ that we observe. We have added this point to the text.

- *Lines 217 - 219: The inferences here are reasonable and potentially important but the statement should be supported by citing relevant work such as that of Chen et al., 2015.*

Done.

- *ADDITIONAL COMMENTS*

1) Warming in fauna and in $d18\text{O}$ after 140 ka is not evident in the methane record, but it may be apparent in the smoothed version of the synthetic Greenland record (Barker's work) and also at nearby high northern latitude sites (Barker et al., 2015) (Mokeddem and McManus, 2016).

Yes. But we think a further discussion of this interval will add unnecessary complication to the overall story.

2) See also Lehman et al. (2002) for relevant SST data from a nearby site.

We believe that the SST record from this core is compatible with our faunal records within the relative uncertainties in both age models. However, without revisiting the Adkins et al. study described earlier (and recalculating their age model, as used by Lehman in this study) we cannot compare the records robustly. This could be done in a future study.

REFERENCES other than work of the authors:

Adkins, J.F., Boyle, E.A., Keigwin, L., Cortijo, E., 1997. Variability of the North Atlantic thermohaline circulation during the last interglacial period. Nature 390, 154-156.

Galaasen, E.V., Ninnemann, U.S., Irvani, N., Kleiven, H.F., Rosenthal, Y., Kissel, C., Hodell, D.A., 2014. Rapid Reductions in North Atlantic Deep Water During the Peak of the Last Interglacial Period. *Science* 343, 1129-1132.

Hodell, D.A., Minth, E.K., Curtis, J.H., McCave, I.N., Hall, I.R., Channell, J.E.T., Xuan, C., 2009. Surface and deep-water hydrography on Gardar Drift (Iceland Basin) during the last interglacial period. *Earth and Planetary Science Letters* 288, 10-19.

Lehman, S.J., Sachs, J.P., Crotwell, A.M., Keigwin, L.D., Boyle, E.A., 2002. Relation of subtropical Atlantic temperature, high-latitude ice rafting, deep water formation, and European climate 130,000- 60,000 years ago. *Quaternary Science Reviews* 21, 1917-1924.

Mokeddem, Z., McManus, J.F., 2016. Persistent climatic and oceanographic oscillations in the subpolar North Atlantic during the MIS 6 glaciation and MIS 5 interglacial. *Paleoceanography*, doi: 10.1002/2015PA002813.

Roberts, N.L., Piotrowski, A.M., McManus, J.F., Keigwin, L.D., 2010. Synchronous deglacial overturning and water mass source changes. *Science* 327, 75-78.

Reviewers' comments:

Reviewer #1 (Remarks to the Author):

Here the authors present a thoroughly revised manuscript. Figures are more thought-out now. Compared with the former version redundant parts of the text have been avoided. The structure is more reasonable. However, I need to notice that the authors did not do a good job in taking into account the most important points of criticism of both reviewers:

Rev2 pointed out: "The principal weakness of the paper is that the reasoning that leads from the data presented by the authors to the main conclusion[...]."

Rev1 mentioned: "[...]I feel they need to focus better on what they conclusively can say from the observations and how to present it."

In this sense the new manuscript did not settle these points of criticism. Instead of focusing on what can be derived from the new observations, the authors now present even more speculative hypotheses.

I'm sorry for being so negative, because I appreciate very much the paleoceanographic inventiveness of the authors. Some of the here presented ideas may possibly have the potential to be discussed in this research field in the following years. In this sense I want to apologize for not being convinced by the here presented line of arguments. But there is too little support for most of their hypotheses. The lines of argumentation are too long. Conclusions are based on too many assumptions and too few observations. Although individual assumptions might be likely reasonable, the long sequence of required assumptions likely leads to false conclusions.

I'm not saying the authors are utterly wrong. But the presented data is certainly not sufficient in order to underpin their far reaching conclusions. The reader still is left alone with a number of hypotheses based on weak evidence, while the conclusions and explanations are vague.

I think the most obvious example for my criticism is the main message in the last sentence of the abstract (and the last figure), which was added to the previous version of the manuscript. This part is totally out of context and not covered by the title as well. I don't see this part supported neither by their new data, nor by any other data. In my opinion it is not an appropriate approach presenting some new data from T2 and extrapolating these observations to four other terminations. This is a highly speculative idea, not even covered by the definition of the authors when looking into details. Line 376: "We define the start of a termination by the onset of (inferred) weak circulation [...]",. However, when looking at figure 5 I see two out of five terminations not showing this behaviour (T5, T3). Thus, this definition seems somehow arbitrary to me. Not mentioning the speculative approach of the *GL_syn_hi* record as an AMOC proxy. AMOC is NOT a synonym for heat transport (Lozier, 2012; Wunsch, 2008, 2010)!

I have the impression that the authors could not decide which message they wanted to focus on. I suggest to split the here presented ideas in at least two papers with support of a model approach in order to explain the mechanisms behind. This would make much more sense than founding this new hypothesis on a motley collection of data of different proxies from different sites.

I recognize that the authors provide an improvement for the age model of ODP1063, but when looking into detail I don't think this is an essential improvement, because the new fish-debris Nd data (Fig. 3) does not sufficiently cover all of T2, in particular the end of the "overshoot". Due to the low time resolution I do not see substantial improvement compared to the Bohm et al. data (24 new data points versus 55 by Bohm in the examined time period of interest). By averaging 5 samples they also lose the temporal resolution, which would have been of interest. Further, I do not understand why they did not produce more Nd data if they consider the leachate approach as not robust enough, which I cannot follow when looking at the high level of concordance between both Nd records. Why should the leaching method not be comparable to fish debris data (Wilson et al., 2013)?

The $\delta^{13}C$ of ODP1063 raises a lot of questions as well. It is obvious that something is wrong here. I agree that these problems most likely are not analytically, instead there might be a systematical problem causing this scatter. Whatever might be the reason for the scatter, this record should not be used as a circulation proxy at this very location.

There are further examples for inaccurate definitions and/or simplified statements about issues, which are presented as common-sense, but which are actually still under debate. References are required in order to underpin such statements.

e.g. Line 50: "speleothems and Antarctic ice cores suggests that the AMOC may have been in a weakened and or shallow mode [...]"- I don't believe that there is such a simple connection.

"deglacial CO₂ change" – likely atmospheric CO₂ is meant? In general the phrase "CO₂ change" is used without definition too often.

line 399: "melting of continental ice sheets can sustain a weakened AMOC until the cumulative release of CO₂ and concomitant degree of deglaciation is sufficient to stabilise an interglacial mode of the AMOC"- what about insolation?

I like the idea that Nd isotopes should be considered in a "NW versus NE+S" manner, rather than simply by "north vs South" problem. But this is not really new and should have set in context to other studies from this area (e.g. (Crocker et al., 2016)). In particular, there is no way to interpret negative Nd signature excursions without discussing the new study by (Howe et al., 2016). In order to provide evidence for this idea there should be measurements of Nd from ODP983 as well. Instead the authors present sortable silt and ¹³C in order to underpin a statement regarding Nd, while both proxies are not in temporal phase with the Nd from ODP1063. I cannot follow this approach.

I also do not consider the concept of an overshoot as new e.g. (Barker et al., 2010; Cheng et al., 2014; Thornalley et al., 2013). Although there has been more and more observational evidence for this phenomenon recently, the here presented data does not support this sufficiently.

References

- Barker, S., G. Knorr, M. Vautravers, P. Diz, and Skinner, L., 2010, Extreme deepening of the Atlantic overturning circulation during deglaciation: *Nature Geoscience*, v. 3.
- Cheng, J., Liu, Z., He, F., Otto-Bliesner, B., Brady, E., and Lynch-Stieglitz, J., 2014, Model-proxy comparison for overshoot phenomenon of Atlantic thermohaline circulation at Bølling-Allerød: *Chinese Science Bulletin*, v. 59, no. 33, p. 4510-4515.
- Crocker, A. J., Chalk, T. B., Bailey, I., Spencer, M. R., Gutjahr, M., Foster, G. L., and Wilson, P. A., 2016, Geochemical response of the mid-depth Northeast Atlantic Ocean to freshwater input during Heinrich events 1 to 4: *Quaternary Science Reviews*, v. 151, p. 236-254.
- Howe, J. N. W., Piotrowski, A. M., and Rennie, V. C. F., 2016, Abyssal origin for the early Holocene pulse of unradiogenic neodymium isotopes in Atlantic seawater: *Geology*.
- Lozier, S., 2012, Overturning in the North Atlantic: *Annual Review of Marine Science*, v. 4, p. 291-315.
- Thornalley, D. J. R., Blaschek, M., Davies, F. J., Praetorius, S., Oppo, D. W., McManus, J. F., Hall, I. R., Kleiven, H., Renssen, H., and McCave, I. N., 2013, Long-term variations in Iceland-Scotland overflow strength during the Holocene: *Climate of the Past*, v. 9, no. 5, p. 2073-2084.
- Wilson, D., A. Piotrowski, A. Galy, and Clegg, J., 2013, Reactivity of neodymium carriers in deep sea sediments: Implications for boundary exchange and paleoceanography: *Geochimica et Cosmochimica Acta*, v. 109, p. 197-221.
- Wunsch, C., 2008, The circulation of the ocean and its variability: *Progress in Physical Geography*, v. 32, no. 4, p. 463-474.
- , 2010, Towards understanding the Paleocean: *Quaternary Science Reviews*, v. 29, no. 17-18, p. 1960-1967.

Reviewer #2 (Remarks to the Author):

As noted in my original review, the new data presented by Deaney et al. will be of broad interest and I look forward to seeing them published. In this revision the authors have provided comprehensive and thoughtful responses to the comments of both referees. I find the presentation

to be much clearer and the interpretation now to be almost fully substantiated. Therefore, I recommend publication with just a few minor changes.

Line 53: Figure 2 is now cited before Figure 1 (Line 68), apparently as a consequence of my comment on the previous version that it would be easier for readers to follow the manuscript if the original Figure 2 (now Figure 1) were presented first. Now the out-of-sequence order of figure citation needs to be corrected. I recommend that the authors simply delete the citation of Figure 2 in Line 53 and replace it with a reference that gives the age of HS11.

Lines 119-122: Here, the inference of "enhanced influence of southern-sourced deep waters" is contingent upon the assumption of constant ϵ_{Nd} composition for each water mass end member throughout the interval of interest. However, as the authors are aware, this assumption is questionable. Rather than adding a long discussion of published evidence for/against variability of water mass ϵ_{Nd} composition, I suggest that the authors simply add the caveat to this statement that the inference is contingent on an assumption of constant water mass ϵ_{Nd} composition.

Line 244: after "of MIS 5e" insert "in the NW Atlantic"

Line 477: Change from to form

Line 471: Change Fig. 2a to Fig. 1a

Line 479: Change Fig. 2b to Fig. 1b

Line 480: Change Fig. 2c to Fig. 1c

Line 494: Change Fig. 1 to Fig. 2

Line 512: Change synchronicity (not a word) to synchronicity

Deaney et al., Response to Reviewers' comments

Reviewers' comments in *italics*, our responses in plain text

Reviewers' comments:

Reviewer #1 (Remarks to the Author):

Here the authors present a thoroughly revised manuscript. Figures are more thought-out now. Compared with the former version redundant parts of the text have been avoided. The structure is more reasonable. However, I need to notice that the authors did not do a good job in taking into account the most important points of criticism of both reviewers:

Rev2 pointed out: "The principal weakness of the paper is that the reasoning that leads from the data presented by the authors to the main conclusion[...]."

Rev1 mentioned: "[...]I feel they need to focus better on what they conclusively can say from the observations and how to present it."

In this sense the new manuscript did not settle these points of criticism. Instead of focusing on what can be derived from the new observations, the authors now present even more speculative hypotheses.

See below – we have removed the section on previous terminations. We believe that we have (now) addressed the concerns of the original Reviewers (as supported by the second-round review of one of those individuals).

I'm sorry for being so negative, because I appreciate very much the paleoceanographic inventiveness of the authors. Some of the here presented ideas may possibly have the potential to be discussed in this research field in the following years. In this sense I want to apologize for not being convinced by the here presented line of arguments. But there is too little support for most of their hypotheses. The lines of argumentation are too long. Conclusions are based on too many assumptions and too few observations. Although individual assumptions might be likely reasonable, the long sequence of required assumptions likely leads to false conclusions.

I'm not saying the authors are utterly wrong. But the presented data is certainly not sufficient in order to underpin their far reaching conclusions. The reader still is left alone with a number of hypotheses based on weak evidence, while the conclusions and explanations are vague.

I think the most obvious example for my criticism is the main message in the last sentence of the abstract (and the last figure), which was added to the previous version of the manuscript. This part is totally out of context and not covered by the title as well. I don't see this part supported neither by their new data, nor by any other data. In my opinion it is not an appropriate approach presenting some new data from T2 and extrapolating these observations to four other terminations. This is a highly speculative idea, not even covered by the definition of the authors when looking into details. Line 376: "We define the start of a termination by the onset of (inferred) weak circulation [...].". However, when looking at figure 5 I see two out of five terminations not showing this behaviour (T5, T3). Thus, this definition seems somehow arbitrary to me. Not mentioning the speculative approach of the GL_syn_hi record as an AMOC proxy. AMOC is NOT a synonym for heat transport

(Lozier, 2012; Wunsch, 2008, 2010)!

I have the impression that the authors could not decide which message they wanted to focus on. I suggest to split the here presented ideas in at least two papers with support of a model approach in order to explain the mechanisms behind. This would make much more sense than founding this new hypothesis on a motley collection of data of different proxies from different sites.

We acknowledge that we may have gone too far with the additional discussion on older terminations. We had envisioned this to be a separate manuscript, and inclusion in the present manuscript was perhaps premature. We have therefore removed this section along with the 'new' Figure 5. We now limit our discussion to the most recent 2 terminations (as in our original submission), for which there is direct evidence for circulation changes and corresponding changes in atmospheric CO₂.

I recognize that the authors provide an improvement for the age model of ODP1063, but when looking into detail I don't think this is an essential improvement, because the new fish-debris Nd data (Fig. 3) does not sufficiently cover all of T2, in particular the end of the "overshoot". Due to the low time resolution I do not see substantial improvement compared to the Bohm et al. data (24 new data points versus 55 by Bohm in the examined time period of interest).

We appreciate the opportunity to clarify where we think our study provides improvement upon the Bohm et al. study. The most significant improvement in our age model is not the difference in absolute age but the temporal constraint provided by our new surface ocean proxy records (isotopes and faunal counts). Without these we would not be able to discuss the relative timing of ocean circulation changes with respect to the ice core data. We would like to emphasise that even if the age model was the same (which it is not), our study would represent an advance because of this constraint. We also provide other proxy indicators that add to our interpretation. For example our new record of benthic $\delta^{18}\text{O}$ provides evidence for the timing (i.e. synchronous with surface warming) and rapidity of deep ocean change that was not available with the more laborious and lower resolution ϵNd reconstructions.

By averaging 5 samples they also loose the temporal resolution, which would have been of interest. Further, I do not understand why they did not produce more Nd data if they consider the leachate approach as not robust enough, which I cannot follow when looking at the high level of concordance between both Nd records. Why should the leaching method not comparable to fish debris data (Wilson et al., 2013)?

We made these Nd isotope measurements as part of a PhD thesis project before we had any knowledge of the Bohm et al study, and only learned about that work when the paper was in press. As for our work, we decided to use fish debris following the study of Roberts et al., (2010), which showed a significant offset between ϵNd measurements obtained on fish debris versus bulk leachates in a core very close to the location of ODP 1063. This was also shown by the Crocker et al., (2016) study at ODP site 980 (see also discussion in Elmore et al. (2011)). Fish debris is often considered the gold standard when using Nd isotopes as a bottom water mass tracer and as such our new measurements provide a very valuable check on the bulk leachate results but they should not be considered merely as a confirmation of the published record. In fact the two records are not

always in agreement and this is why we have not combined the two. As pointed out in the original reviews, the fish debris data were very hard won and they are at the maximum resolution possible for the interval given the low abundance of this material in marine sediment such as this. We did not average 5 samples, rather we combined them just in order to collect enough material to analyse. This explains why our record has lower resolution than the Bohm record. We would like to emphasise that we are not trying to sell our study based on 'better' Nd isotope data, but rather on the fact that we present several new parallel records, reflecting both surface and deep water properties, which allow us to go beyond the Bohm et al study.

The $\delta^{13}\text{C}$ of ODP1063 raises a lot of questions as well. It is obvious that something is wrong here. I agree that these problems most likely are not analytically, instead there might be a systematical problem causing this scatter. Whatever might be the reason for the scatter, this record should not be used as a circulation proxy at this very location.

We would like to keep these records while making the explicit caveat that the scatter is difficult to explain. We employ the benthic $\delta^{13}\text{C}$ results in combination with indices for carbonate preservation to emphasise the increase in deep ocean ventilation $\sim 129\text{ka}$, and we note that the $\delta^{13}\text{C}$ record shows a clear deglacial signal even without smoothing. We also note that recent studies (including Poirier and Billups, 2014, *Paleoceanography* 10.1002/2014PA002661) based their conclusions on ocean circulation solely on benthic $\delta^{13}\text{C}$ from this site. We also would like to point out that other sites also seem to be affected by this phenomenon. For example the benthic $\delta^{13}\text{C}$ record from U1304 (published by Hodell et al., *EPSL*, 288, p10, 2009 and used as the only circulation tracer at that site) has a 5 point running mean applied to an equally scattered record.

There are further examples for inaccurate definitions and/or simplified statements about issues, which are presented as common-sense, but which are actually still under debate. References are required in order to underpin such statements.

e.g. Line 50: "speleothems and Antarctic ice cores suggests that the AMOC may have been in a weakened and or shallow mode [...]" - I don't believe that there is such a simple connection.

As the Reviewer points out, the connection is probably not simple, but has been implied by previous studies (and many times by the speleothem community e.g. Cheng et al., *Science*, 326, p248, 2009; Cheng et al., *Nature*, 534, p640, 2016). As such we have amended this sentence to "By analogy with similar conditions associated with Heinrich Stadial 1 (HS1, $\sim 18\text{-}14.6\text{ka}$) during the early part of T1, evidence from North Atlantic marine sediments, Chinese speleothems and Antarctic ice cores **has been used to infer** that the AMOC may have been in a weakened and or shallow mode throughout much of T2 [Cheng et al., 2009; Barker et al., 2011] "

"deglacial CO2 change" – likely atmospheric CO2 is meant? In general the phrase "CO2 change" is used without definition too often.

We have added more references to 'atmospheric' throughout the text.

line 399: "melting of continental ice sheets can sustain a weakened AMOC until the cumulative release of CO2 and concomitant degree of deglaciation is sufficient to stabilise an interglacial mode of the AMOC"- what about insolation?

Insolation can play a role in the melting of ice sheets, which we mentioned in line 394 of the previous version of the manuscript. We have now removed this entire section but we do acknowledge the influence of insolation on line 397 of the revised text.

I like the idea that Nd isotopes should be considered in a “NW versus NE+S” manner, rather than simply by “north vs South” problem. But this is not really new and should have set in context to other studies from this area (e.g. (Crocker et al., 2016)). In particular, there is no way to interpret negative Nd signature excursions without discussing the new study by (Howe et al., 2016).

Neither the Crocker nor Howe studies were published when we submitted our manuscript. In fact, one of the present authors published a paper recently that also makes this statement based on modern water column measurements and points out the significance for paleoceanographic studies (van de Flierdt et al., 2016, Proceed. Phil. Trans. R. Soc.) We agree however that the Howe et al. study is relevant to our findings and have added the following to the revised manuscript (see Line 194):

“Similarly negative ϵNd values have been documented at the location of ODP site 1063 during the early Holocene [Roberts et al., 2010; Howe et al., 2016]. In particular, Howe et al., [Howe et al., 2016] conclude that these very negative early Holocene ϵNd values might reflect the ‘re-labelling’ of deep waters in the Labrador Sea by interaction with particularly un-radiogenic sediments, followed by their southward advection to abyssal depths. A key question remains as to whether these deep waters originated in the NW or NE Atlantic..”

In order to provide evidence for this idea there should be measurements of Nd from ODP983 as well. Instead the authors present sortable silt and ^{13}C in order to underpin a statement regarding Nd, while both proxies are not in temporal phase with the Nd from ODP1063. I cannot follow this approach.

We extended the discussion of deep water formation in the NE versus NW Atlantic in response to one of the original Reviewers’ comments concerning previous studies by Hodell et al (2009) and Galaasen et al., (2014) that provided evidence for the later ($\sim 124\text{ka}$) recovery of a typical interglacial mode of AMOC. We wanted to highlight that our results are compatible with those studies when considering that the sites they used are particularly sensitive to deep waters forming in the NE Atlantic (Nordic Seas). The site of ODP 983 is directly downstream from Iceland and previous studies have warned against the application of Nd isotopes in this region, which is heavily influenced by volcanic input (e.g. Elmore et al., 2011, G3, 10.1029/2011GC003741; Roberts and Piotrowski, 2015). Therefore we used Sortable Silt and benthic $\delta^{13}\text{C}$ to assess the change in deep water overflows emanating from the Nordic Seas. Our argument is that deep waters originating in the NE Atlantic were ‘less dense’ with respect to e.g. deep waters from the NW Atlantic prior to $\sim 124\text{ka}$ and that this situation changed $\sim 124\text{ka}$ with the deepening of the Nordic Seas overflows to form more modern-like LNADW. This explains why the circulation tracers for 983 are not in phase with those from 1063. ODP 983 is more sensitive to the Nordic Seas overflows, which deepen $\sim 124\text{ka}$ whereas ODP 1063 records an earlier change, which was therefore presumably not from the Nordic Seas.

We therefore think that the additional data and discussion of ODP 983 adds significant value to our study, providing a possible reconciliation with previous studies.

I also do not consider the concept of an overshoot as new e.g. (Barker et al., 2010; Cheng et al., 2014; Thornalley et al., 2013). Although there has been more and more observational evidence for this phenomenon recently, the here presented data does not support this sufficiently.

We did not intend to suggest that the concept of an overshoot is new. In contrast, we refer to previous studies concerning an overshoot during Termination 1 (e.g. Barker et al., 2010 and Liu et al., 2009). We agree that our evidence for an overshoot following HS 11 is by analogy with observations from the most recent deglaciation and therefore referred to it as our 'inferred overshoot' in our original text. We have gone back through the text and added more instances of 'inferred' to the revised text so as not to overstate our interpretation

Reviewer #2 (Remarks to the Author):

As noted in my original review, the new data presented by Deaney et al. will be of broad interest and I look forward to seeing them published. In this revision the authors have provided comprehensive and thoughtful responses to the comments of both referees. I find the presentation to be much clearer and the interpretation now to be almost fully substantiated. Therefore, I recommend publication with just a few minor changes.

Line 53: Figure 2 is now cited before Figure 1 (Line 68), apparently as a consequence of my comment on the previous version that it would be easier for readers to follow the manuscript if the original Figure 2 (now Figure 1) were presented first. Now the out-of-sequence order of figure citation needs to be corrected. I recommend that the authors simply delete the citation of Figure 2 in Line 53 and replace it with a reference that gives the age of HS11.

Done

Lines 119-122: Here, the inference of "enhanced influence of southern-sourced deep waters" is contingent upon the assumption of constant $\epsilon\text{-Nd}$ composition for each water mass end member throughout the interval of interest. However, as the authors are aware, this assumption is questionable. Rather than adding a long discussion of published evidence for/against variability of water mass $\epsilon\text{-Nd}$ composition, I suggest that the authors simply add the caveat to this statement that the inference is contingent on an assumption of constant water mass $\epsilon\text{-Nd}$ composition.

We have added the caveat:

"the record of ϵNd from ODP 1063 cannot be interpreted as a simple proxy for the mixing ratio between northern and southern deep water sources **even if the ϵNd composition of the various deep water end-members remained constant.**"

Line 244: after "of MIS 5e" insert "in the NW Atlantic"

Done

Line 477: Change from to form

This was not actually a typo, the sentence refers to a subset of samples from the northern part of the GEOTRACES transect GA02

Line 471: Change Fig. 2a to Fig. 1a

Done

Line 479: Change Fig. 2b to Fig. 1b

Done

Line 480: Change Fig. 2c to Fig. 1c

Done

Line 494: Change Fig. 1 to Fig. 2

Done

Line 512: Change synchronicity (not a word) to synchronicity

Happy to change (although synchronicity is a word and is a synonym of synchronicity!)

EVIEWERS' COMMENTS:

Reviewer #1 (Remarks to the Author):

Improved by two rounds of reviews the manuscript is now back to the stage before submission two. I'm glad the authors removed the parts on the precursor terminations.

Although I appreciate the author's efforts and innovativeness I still consider the presented evidence not as extremely robust. Again, I'm not saying that the authors are wrong. I just think the lines of argumentation are still too long to sufficiently support the conclusions drawn here.

Example:

line 227: "While the record of SS from a single water depth cannot tell us about the gross flux of ISOW it is nevertheless instructive."

In the following sentences the authors explain why this single record is still valuable and that it is embedded in a larger data set. The larger data set is thus used as the justification of using a single core site. I wonder why not using then the larger data set right away?

The same with the $\delta^{18}O$ from ODP1063 (line 253 onwards): here $\delta^{18}O$ is used as a proxy for water mass geometry, which is new to me. Again, I'm not saying the authors are wrong. But here again they need a number of sentences to explain why this could be right, while I would simply expect additional support for this conclusion from more other locations here. There is plenty available.

So the fundamental problem of this manuscript is that (almost) global conclusions are drawn from the single sediment core ODP1063 (there is also ODP983, but this core is not even shown in Figure 1c).

line 236: it is found that the SS record of ODP983 is not in line with the Nd-record of ODP1063. The increase of flow speed starts ~5 kyrs later at 983. Again the authors need a number of sentences to explain that. They need to use terms like "could explain" and "may", which is a correct approach here (and indicates that the authors are not overselling something), but it also indicates, that there is a non-negligible amount of speculation. Thus, I wonder if ODP983 is helpful at all in supporting the main finding from the abstract, which is: "Here we provide multiple lines of evidence suggesting that the ~20ppmv overshoot in CO₂ at the end of Termination 2 (T2) ~129ka was associated with an abrupt (≤ 400 yr) deepening of Atlantic Meridional Overturning Circulation (AMOC)".

However, I don't want to hamper the publication of this manuscript if the authors and the editor believe that the here presented lines of arguments are sufficient for Nature Communications to support the conclusions. The methods are technically sound and the writing is excellent (although the figures still have some redundancy). In my personal opinion I would have expected a compilation of stacked proxy data from well-selected sites, independent of how hard-won the own data is.

This new submission did not come along with any supplement. I assume it did not change from last time?

Reviewer #1 (Remarks to the Author):

Improved by two rounds of reviews the manuscript is now back to the stage before submission two. I'm glad the authors removed the parts on the precursor terminations. Although I appreciate the author's efforts and innovativeness I still consider the presented evidence not as extremely robust. Again, I'm not saying that the authors are wrong. I just think the lines of argumentation are still too long to sufficiently support the conclusions drawn here.

Example:

line 227: "While the record of SS from a single water depth cannot tell us about the gross flux of ISOW it is nevertheless instructive."

In the following sentences the authors explain why this single record is still valuable and that it is embedded in a larger data set. The larger data set is thus used as the justification of using a single core site. I wonder why not using then the larger data set right away?

The same with the d18O from ODP1063 (line 253 onwards): here d81O is used as a proxy for water mass geometry, which is new to me. Again, I'm not saying the authors are wrong. But here again they need a number of sentences to explain why this could be right, while I would simply expect additional support for this conclusion from more other locations here. There is plenty available.

So the fundamental problem of this manuscript is that (almost) global conclusions are drawn from the single sediment core ODP1063 (there is also ODP983, but this core is not even shown in Figure 1c).

line 236: it is found that the SS record of ODP983 is not in line with the Nd-record of ODP1063. The increase of flow speed starts ~5 kyrs later at 983. Again the authors need a number of sentences to explain that. They need to use terms like "could explain" and "may", which is a correct approach here (and indicates that the authors are not overselling something), but it also indicates, that there is a non-negligible amount of speculation. Thus, I wonder if ODP983 is helpful at all in supporting the main finding from the abstract, which is: "Here we provide multiple lines of evidence suggesting that the ~20ppmv overshoot in CO₂ at the end of Termination 2 (T2) ~129ka was associated with an abrupt (≤ 400 yr) deepening of Atlantic Meridional Overturning Circulation (AMOC)".

However, I don't want to hamper the publication of this manuscript if the authors and the editor believe that the here presented lines of arguments are sufficient for Nature Communications to support the conclusions. The methods are technically sound and the writing is excellent (although the figures still have some redundancy). In my personal opinion I would have expected a compilation of stacked proxy data from well-selected sites, independent of how hard-won the own data is.

This new submission did not come along with any supplement. I assume it did not change from last time?

Author response

We feel that our arguments are as long as is required to explain our reasoning sufficiently. We do not agree that a long argument is necessarily a speculative one although the Reviewer is correct in that we do not wish to oversell our interpretations.

We are not aware of any other published records of ϵNd that could support our conclusions and addition of other datasets (e.g. benthic $\delta^{18}\text{O}$) from other core sites would in general require their alignment to ours, thus limiting their support as independent records.

The wider sortable silt dataset referred to by the Reviewer exists only for the Early Holocene, not across T2.

We have added text (see below) to make it clear that our conclusions are based on a limited number of core sites and that future work needs to be done on other sites to test our conclusions.

Line 246: "...acknowledging the limitations of a single core site"

Line 394: "We also acknowledge that the majority of our discussion is based on findings from a single core site in the NW Atlantic. Future validation of our results will require equivalent reconstructions from a variety of sites across a much broader region."